# ROCO: A General Framework for Evaluating Robustness of Combinatorial Optimization Solvers on Graphs

**Han Lu**[1#]**, Zenan Li**[1#]**, Runzhong Wang**[1]**, Qibing Ren**[1]**, Xijun Li**[2]**, Mingxuan Yuan**[2]**,
Jia Zeng**[2]**, Xiaokang Yang**[1]**, Junchi Yan**[1*]
[1]MoE Key Lab of Artificial Intelligence, Shanghai Jiao Tong University      [2]Huawei Noah's Ark Lab
`{sjtu_luhan,emiyali,runzhong.wang,renqibing,xkyang,yanjunchi}@sjtu.edu.cn`
`{xijun.li,yuan.mingxuan,zeng.jia}@huawei.com`
Code:  `https://github.com/Thinklab-SJTU/ROCO`

## Abstract

Solving combinatorial optimization (CO) on graphs has been attracting increasing interests from the machine learning community whereby data-driven approaches were recently devised to go beyond traditional manually-designated algorithms. In this paper, we study the robustness of a combinatorial solver as a blackbox regardless it is classic or learning-based though the latter can often be more interesting to the ML community. Specifically, we develop a practically feasible robustness metric for general CO solvers. A no-worse optimal cost guarantee is developed as such the optimal solutions are not required to achieve for solvers, and we tackle the non-differentiable challenge in input instance disturbance by resorting to black-box adversarial attack methods. Extensive experiments are conducted on 14 unique combinations of solvers and CO problems, and we demonstrate that the performance of state-of-the-art solvers like Gurobi can degenerate by over 20% under the given time limit bound on the hard instances discovered by our robustness metric, raising concerns about the robustness of combinatorial optimization solvers.

## 1 Introduction

The combinatorial optimization (CO) problems on graphs are widely studied due to their important applications including aligning cross-modality labels (Lyu et al., 2020), discovering vital seed users in social networks (Zhu et al., 2019), tackling graph matching problems (Wang et al., 2020; 2022) and scheduling jobs in data centers (Mao et al., 2019), etc. However, CO problems are non-trivial to solve due to the NP-hard challenge, whereby the optimal solution can be nearly infeasible to achieve for even medium-sized problems. Existing approaches to practically tackle CO include heuristic methods (Van Laarhoven & Aarts, 1987; Whitley, 1994), powerful branch-and-bound solvers (Gurobi Optimization, 2020; The SCIP Optimization Suite 8.0, 2021; Forrest et al., 2022) and recently developed learning-based models (Khalil et al., 2017; Yu et al., 2020; Kwon et al., 2021).

Despite the success of solvers in various combinatorial tasks, little attention has been paid to the vulnerability and robustness of combinatorial solvers. As pointed out by Yehuda et al. (2020), we cannot teach a perfect solver to predict satisfying results for all input CO problems. Within the scope of the solvers and problems studied in this paper, our results shows that the performance of the solver may degenerate a lot given certain data distributions that should lead to the same or better solutions compared to the original distribution assuming the solver works robustly. We also validate in experiments that such a performance degradation is neither caused by the inherent discrete nature of CO. Such a discovery raises our concerns about the robustness (i.e. the capability to perform stably w.r.t. perturbations on problem instances) of combinatorial solvers, which is also aware by Varma & Yoshida (2021); Geisler et al. (2021). However, Varma & Yoshida (2021) focuses on theoretical analysis and requires the optimal solution, which is infeasible to reach in practice. Geisler et al. (2021)

---

[*]Han Lu and Zenan Li contribute equally. Junchi Yan is the correspondence author who is also with Shanghai AI Laboratory. The work was in part supported by National Key Research and Development Program of China (2020AAA0107600), NSFC (62222607, 72192821) and STCSM (22511105100).

Table 1: We make a (no strict) analogy by comparing the proposed ROCO framework with two popular attacking paradigms FGSM (Goodfellow et al., 2015) and RL-S2V (Dai et al., 2018). $\epsilon$-perturb means the change of one pixel should be bounded in $\epsilon$. B-hop neighbourhood means the new attack edges can only connect two nodes with distance less than $B$.

| Method | Data | Task | Attack target | Attack budget | Attack principle |
|---:|:---:|:---:|:---:|:---:|:---:|
| FGSM | image | classification | pixels | $\epsilon$-Perturb | invisible change |
| RL-S2V | graph | classification | edges (connectivity) | # edge | B-hop neighbour |
| ROCO (ours) | CO instance | solving CO | edges (constraints/cost) | # edge | no worse optimum |

applies the attack-and-defense study in deep learning into learning-based combinatorial optimization models, requiring a differentiable neural network that is infeasible for general solvers that can often be undifferentiable. Our paper also differs from the namely "robust optimization" (Buchheim & Kurtz, 2018), where the expected objective score is optimized given known data distribution. We summarize the challenges and our initiatives of evaluating the robustness of existing solvers as follows:

**1) Robustness metric without optimal solutions**: The underlying NP-hard challenge of combinatorial optimization prohibits us from obtaining the optimal solutions. However, the robustness metric proposed by Varma & Yoshida (2021) requires optimal solutions, making this metric infeasible in practice. In this paper, we propose a problem modification based robustness evaluation method, whereby the solvers' performances are evaluated on problems that can guarantee *no worse optimal costs*. That is, the optimal solution of the modified problem instance gives better objective value than the original problem, which resolves the needs of the optimal solution.

**2) Robustness metric of general (non-differentiable) solvers**: Despite the recently developed deep-learning solvers, most existing solvers are non-differentiable due to the discrete combinatorial nature. Thus, the gradient-based robustness metric by Geisler et al. (2021) has the limitation to generalize to general solvers. In this paper, we develop a reinforcement learning (RL) based attacker that modifies the CO problems to get hard problem instances (where the solvers perform poorly) with no worse optimal costs. Regarding arbitrary types of solvers as black-boxes, the RL agent is trained by policy gradient without requiring the solver to be differentiable. Our framework also owns the flexibility where the agent can be replaced by other search schemes e.g. simulated annealing.

Being aware that many CO problems can be essentially formulated as a graph problem and there are well-developed graph-based learning models (Scarselli et al., 2008; Kipf & Welling, 2016; Velickovic et al., 2017), the scope of this paper is restricted within combinatorial problems on graphs following (Khalil et al., 2017; Wang et al., 2021). To develop a problem modification method that achieves *no worse optimal costs*, we propose to modify the graph structure with a problem-dependent strategy exploiting the underlying problem natures. Two strategies are developed concerning the studied four combinatorial problems: 1) loosening the constraints such that the feasible space can be enlarged while the original optimal solution is preserved; 2) lowering the cost of a partial problem such that the objective value of the original optimal solution can become better. Such strategies are feasible when certain modifications are performed on the edges, and tackle the challenge that we have no access to the optimal solutions.The generated data distributions can also be close to the original data by restricting the number of edges modified. The graph modification steps are performed by our proposed attackers that generate problem instances where the solvers' performances degenerate, and we ensure the *no worse optimal costs* guarantee by restricting the action space of the attackers.

Our framework can be seen as the attack on the solvers and how much the worst cases near the clean instances will do harm to the solver are considered as the robustness. Tbl. 1 compares our framework to classical works on adversarial attacks for images/graphs. **The highlights of this paper are:**

**1)** We propose **Ro**bust **C**ombnaotorial **O**ptimization (ROCO), a general framework (see Fig. 1) to empirically evaluate the robustness of CO solvers on graphs. It is performed without requiring optimal value or differentiable property.

**2)** We design a novel robustness evaluation method with *no worse optimal costs* guarantee to eliminate the urgent requirements for the optimal solution of CO problems. Specifically, we develop a reinforcement learning (RL) based attacker combined with other search-based attackers to regard the solvers as black-boxes, even they are non-differentiable solvers.

**3)** Comprehensive case studies are conducted on four common combinatorial tasks: Directed Acyclic Graph Scheduling, Asymmetric Traveling Salesman Problem, Maximum Coverage, and Maximum

Coverage with Separate Coverage Constraint, with three kinds of solvers: traditional solvers, learning-based solver and specific MILP solver like Gurobi.

The results presented in this paper may raise the community's attention to the robustness, e.g. big performance drop for the commercial solver Gurobi though it is not a big surprise given the NP-hard nature of many CO problems. It can also be connected to data augmentation for learning-based CO solvers or other methods like parameter tuning to improve solvers' robustness (see Appendix K for more results and discussions). We leave more advanced defense strategies for future work which is still an open problem (Yehuda et al., 2020).

**Remark.** Note we somewhat 'abuse' the term 'attack' which in fact originates from the area of adversarial attack/defense e.g. Goodfellow et al. (2015). Here we use it to express the concept of hard case mining on 'clean' (i.e. raw) instances. In fact, we try to make an analogy between the attacks for image data and our case while we are also aware there are fundamental differences. To ease readers' potential confusion and make the presentation more precise, we will give the formal definition of attack in our context for CO solvers' robustness evaluation in Def. 1 in Sec. 2.1.

The related work is deferred to Appendix A, which mainly refers to the solvers for combinatorial optimization on graphs including traditional and recent learning-based solvers, as well as adversarial attack and defense especially for graph learning.

## 2 THE ROCO FRAMEWORK

### 2.1 ADVERSARIAL ROBUSTNESS FOR CO

**Formulation.** In general, a CO problem aims to find the optimal solution under a set of constraints (usually encoded by a latent graph). Here we formulate a CO problem $Q$ as:

$$\min_{\mathbf{x}} c(\mathbf{x}, Q) \ \ s.t. \ \ h_i(\mathbf{x}, Q) \leq 0, \ i = 1, ..., I \tag{1}$$

where $\mathbf{x}$ denotes the decision variable (i.e. solution) that should be discrete, $c(\mathbf{x}, Q)$ denotes the cost function given problem instance $Q$ and $\{h_i(\mathbf{x}, Q)\}_{i=1}^{I}$ represents the set of constraints. For example, in DAG scheduling, the constraints ensure that the solution $\mathbf{x}$, i.e. the execution order of the DAG job nodes, lies in the feasible space and does not conflict the topological dependency structure of $Q$. However, due to the NP-hard nature, the optimal solution $\mathbf{x}^*$ can be intractable within polynomial time. Therefore, we use a solver $\hat{\mathbf{x}} = f_\theta(Q)$, which gives a mapping $f_\theta : \mathbb{Q} \to \mathbb{X}$ to *approximate* the optimal solution $\mathbf{x}^*$. In this work, $\theta$ are solvers' parameters (e.g. weights and biases in a neural solver, hyperparameters in Gurobi), $Q \in \mathbb{Q}$ is the problem instance, and $\hat{\mathbf{x}} \in \mathbb{X}$ is the approximated solution. $\hat{\mathbf{x}}$ also lies in the feasible space that satisfies all the constraints.

**Solvers' robustness.** In this paper, we would like to raise the concern that the estimation capability of the solvers can be unstable and fail on certain hard instances. Thus, we need methods to discover the non-trivial hard instances for the solvers as the robustness metric where a more robust solver can perform better on hard instances. Here non-trivial means these hard instances are not obtained by trivial operations like increasing the size of problems. In short, given a certain solver $f_\theta$, *we design an attacker (i.e. perturbation model) g to modify the existing clean instances $\tilde{Q} = g(f_\theta, Q)$ and discover hard instances around.* For example, the attacker can modify a limited number of constraints in $Q$ to get a new problem $\tilde{Q}$. Besides working as a robustness metric, the discovered hard instances can guide the parameter setting of solvers, i.e. design solvers' parameters $\theta$ for better performance (robustness) on hard instances, which is beyond the scope of this paper and we mainly focus on developing the first practically feasible robustness metric. Some preliminary results on hard case utilization are presented in Appendix K.

**Robustness metric.** A natural evaluation of the solvers' robustness is to use the gap $c(\hat{\mathbf{x}}, Q) - c(\mathbf{x}^*, Q)$ (Varma & Yoshida, 2021), where a narrower gap stands for a better performance. However, the optimum $\mathbf{x}^*$ is usually unavailable due to the NP-hardness. In this paper, we propose a robustness evaluation method without requiring the optimal solution, and the attack is defined by:

**Definition 1** (Successful Attack). *Given a solver $f_\theta$ and a clean CO problem instance $Q$, we obtain a new problem $\tilde{Q} = g(f_\theta, Q)$ by the attacker g s.t. the optimal cost value after perturbation will become no-worse, i.e. $c(\tilde{\mathbf{x}}^*, \tilde{Q}) \leq c(\mathbf{x}^*, Q)$, where $\tilde{\mathbf{x}}^*$ denotes the new optimal solution. A successful attack occurs when $c(f_\theta(\tilde{Q}), \tilde{Q}) > c(f_\theta(Q), Q)$.*

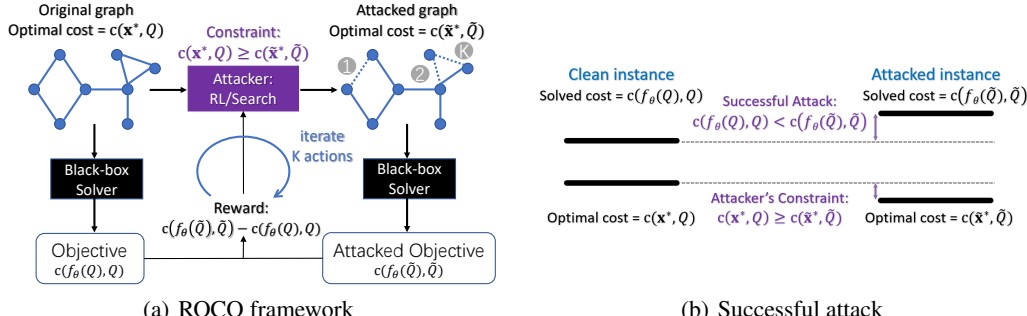

(a) ROCO framework                    (b) Successful attack

Figure 1: Overview and intuition of our attack framework. **Left:** Overview of our attack framework ROCO for CO solvers. ROCO targets on the CO problems which can be encoded by graphs (often holds in practice). Here modifying the edges in the encoded graph represents modifying constraints or lowering the cost in CO. **Right:** The definition of a successful attack.

The intuition is that we restrict the attacker to discover a new problem instance with no-worse optimal cost value, such that the gap between the solver and the optimal will be definitely enlarged if the solver gives a worse cost value:

$$c(\tilde{\mathbf{x}}^*, \tilde{Q}) \leq c(\mathbf{x}^*, Q), \; c(f_\theta(\tilde{Q}), \tilde{Q}) > c(f_\theta(Q), Q) \Rightarrow$$
$$c(f_\theta(Q), Q) - c(\mathbf{x}^*, Q) < c(f_\theta(\tilde{Q}), \tilde{Q}) - c(\tilde{\mathbf{x}}^*, \tilde{Q}), \tag{2}$$

which represents a successful attack (see Fig. 1(b)).

**Attacker's target.** In conclusion, the attacker should maximize the solver's cost value after perturbation while conforming to the *no-worse* restriction. Therefore, we formulate our attack objective:

$$\max_{\tilde{Q}} \quad c(f_\theta(\tilde{Q}), \tilde{Q}) - c(f_\theta(Q), Q)$$
$$s.t. \quad \tilde{Q} = g(f_\theta, Q), \; c(\tilde{\mathbf{x}}^*, \tilde{Q}) \leq c(\mathbf{x}^*, Q) \tag{3}$$

As a consensus in the filed of robustness, the robustness definition should be against the attacker's objective (Carlini et al., 2019; Chen et al., 2022a). Now with the attack objective, we can successively define solvers' robustness quantitatively:

**Definition 2** (Solvers' Robustness). *Given a certain solver $f_\theta$ on a test dataset $\mathcal{D}$ consisted of CO problems, its robustness $r$ is defined as the inverse of the expected worst case cost value after attacked by the attacker $g$ under the no worse optimal cost constraint:*

$$r = -\mathbb{E}_{Q \sim \mathcal{D}}[\max_{\tilde{Q}} c(f_\theta(\tilde{Q}), \tilde{Q})]$$
$$s.t. \quad \tilde{Q} = g(f_\theta, Q), \; c(\tilde{\mathbf{x}}^*, \tilde{Q}) \leq c(\mathbf{x}^*, Q) \tag{4}$$

We add the negative sign in Eq. 4 so that a higher value denotes better robustness.

**Attacker's constraints.** Two types of strategies are developed to meet the *no worse optimal cost* constraint for the attackers: **1)** Loosening the constraints of the problem. In this way, the original optimal solution still lies in the feasible space and its cost is not affected, while the enlarged feasible space may include better solutions. Therefore, the new optimal cost cannot be worse than the original one, and we discover this strategy effective for DAG scheduling (see Section 3.1). **2)** Modifying a partial problem by lowering the costs. Such a strategy ensures that the cost that corresponds to the original optimal solution can never become worse, and there are also chances to introduce solutions with lower costs. We demonstrate its successful application in traveling salesman (see Section 3.2) and max coverage problems (see Section 3.3 & Appendix C).

**Attack via graph modification.** As mentioned above, a CO problem $Q$ can usually be encoded by a latent graph $\mathcal{G} = (V, E)$. For example, the DAG problem can be represented by a directed acyclic graph (see Fig. 4), s.t. the graph nodes stand for the DAG jobs, the directed edges denote the sequential constraints for the jobs, and the node features represent the jobs' completion time. Hence, the perturbation to the problem instance can be conducted by modifications to the graph structure (Dai et al., 2018). Besides, to get non-trivial hard instances around clean examples, the attack is limited within a given attack budget $K$, e.g. the attacker can remove $\leq K$ edges in a DAG problem $Q$. Typically, in our setting, the graph modification is all conducted by *modifying edges*.

## 2.2 IMPLEMENTATIONS OF THE ATTACKER

**Random attack baseline.** The baseline simply chooses and modifies $K$ random edges sequentially, then calculates the new cost. We compare other attackers (which achieve much higher attack effects) to the random attack baseline to claim that *our framework can uncover far more hard instances than the inherent instability caused by the CO problems*.

**Reinforcement learning attacker** *(ROCO-RL)*. We use Eq. 3 as the objective and resort to reinforcement learning (RL) to find $\tilde{Q}$ in a data-driven manner. Specifically, we modify the graph structure and compute $c(f_\theta(\tilde{Q}), \tilde{Q})$ alternatively, getting rewards that will be fed into the PPO (Schulman et al., 2017) framework and train the agent.

Given an instance $(f_\theta, Q)$ along with a modification budget $K$, we model the attack via sequential edge modification as a Markov Decision Process (MDP):

*State.* The current problem instance $Q^k$ (i.e. the problem instance $\tilde{Q}$ obtained after $k$ actions) is treated as the state. The original problem instance $Q^0 = Q$ is the starting state.

*Action.* The attacker is allowed to modify edges in the graph. So a single action at time step $k$ is $a^k \in \mathcal{A}^k \subseteq E^k$. Here our action space $\mathcal{A}^k$ is usually a subset of all the edges $E^k$ since we can narrow the action space (i.e. abandon some useless edge candidates) according to the previous solution $f_\theta(Q^k)$ to speed up our algorithm. Furthermore, we decompose the action space $(O(|V|^2) \to O(|V|))$ by transforming the edge selection into two sequential node selections: first selecting the start node, then the end node.

*Reward.* The objective of the new CO problem $Q^{k+1}$ is $c(f_\theta(Q^{k+1}), Q^{k+1})$. To maximize the new cost value, we define the reward as the increase of the objective:

$$r = c(f_\theta(Q^{k+1}), Q^{k+1}) - c(f_\theta(Q^k), Q^k) \tag{5}$$

*Terminal.* Once the agent modifies $K$ edges or edge candidates become empty, the process stops.

The input and constraints of a CO problem can often be encoded in a graph $\mathcal{G} = (V, E)$, and the PPO agent (i.e. the actor and the critic) should behave according to the graph features. We resort to the Graph Neural Networks (GNNs) for graph embedding:

$$\mathbf{n} = \text{GNN}(\mathcal{G}^k), \quad \mathbf{g} = \text{AttPool}(\mathbf{n}) \tag{6}$$

where $\mathcal{G}^k$ is the corresponding graph of problem $Q^k$, the matrix $\mathbf{n}$ (with the size of node number $\times$ embedding dim) is the node embedding. An attention pooling is used to extract a graph level embedding $\mathbf{g}$. The GNN model can differ by the CO problems (details in Appendix D). After graph feature extraction, we design the actor and critic nets:

*Critic.* It predicts the value of each state $Q^k$. Since it aims at reward maximization, a max-pooling layer is adopted over all node features which are concatenated (denoted by $[\cdot||\cdot]$) with the graph embedding $\mathbf{g}$, fed into a network (e.g. ResNet block (He et al., 2016)) for value prediction:

$$\mathcal{V}(Q^k) = \text{ResNet}_1([\text{MaxPool}(\mathbf{n})||\mathbf{g}]) \tag{7}$$

*Actor.* The edge selection is implemented by selecting the start and end node sequentially. The action scores are computed using two independent ResNet blocks, and a Softmax layer is added to regularize the scores into probability $[0, 1]$:

$$\begin{aligned} P(a_1) &= \text{Softmax}(\text{ResNet}_2([\mathbf{n}||\mathbf{g}])), \\ P(a_2|a_1) &= \text{Softmax}(\text{ResNet}_3([\mathbf{n}||\mathbf{n}[a_1]||\mathbf{g}])) \end{aligned} \tag{8}$$

where $\mathbf{n}[a_1]$ denotes node $a_1$'s embedding. We add the feature vector of the selected start node for the end node selection. For training, actions are sampled by their probabilities. For testing, beam search is used to find the optimal solution: actions with top-$B$ probabilities are chosen for each graph in the last time step. Only those actions with top-$B$ rewards will be reserved for the next search step (see Alg. 1).

**Other attacker implementations.** We also implement three traditional attack algorithms for comparison: random search, optimum-guided search and simulated annealing.

*i) Random Search (ROCO-RA):* In each iteration, an edge is randomly chosen to be modified in the graph and it will be repeated for $K$ iterations. Different from the random attack baseline without any

search process, we will run $N$ attack trials and choose the best solution. Despite its simplicity, it can reflect the robustness of solvers with the cost of time complexity $O(NK)$.

*ii) Optimum-Guided Search (ROCO-OG):* This method focuses on finding the optimum solution during each iteration. We use beam search to maintain the best $B$ current states and randomly sample $M$ different actions from the candidates to generate next states. The number of iterations is set to be no more than $K$. The time complexity is $O(BMK)$.

*iii) Simulated Annealing (ROCO-SA):* Simulated annealing (Van Laarhoven & Aarts, 1987) comes from the idea of annealing and cooling used in physics for particle crystallization. In our scenario, a higher temperature indicates a higher probability of accepting a worse solution, allowing to jump out of the local optimum. As the action number increases, the temperature decreased, and we tend to reject bad solutions. The detailed process is shown in Appendix B and we will repeat it for $N$ times. SA is a fine-tuned algorithm and we can use grid search to find the best parameter to fit the training set. Its time complexity is $O(NMK)$.

Tbl. 7 in Appendix I concludes the attackers' property and complexity. Since the three traditional algorithms are inherently stochastic, we run them multiple times. The trained RL attacker behaves deterministicly and we choose the best trained model.

---

**Algorithm 1: ROCO: Attack via iterative edge manipulation on the input graph**

**Input:** Input problem $Q$; solver $f_\theta$; max number of actions $K$; beam size $B$.
$Q_{1..B}^0 \leftarrow Q$; $\tilde{Q} \leftarrow Q$; # set initial state
**for** $k \leftarrow 1..K$ **do**
  **for** $b \leftarrow 1..B$ **do**
    # do beam search for problems in last step
    Predict $P(a_1)$, $P(a_2|a_1)$ on $Q_b^{k-1}$;
    Select $(a_1, a_2)$ with top-$B$ probabilities;
  **for** each $(b, a_1, a_2)$ pair **do**
    $Q'(b, a_1, a_2) \leftarrow$ modify edge $(a_1, a_2)$ in $Q_b^{k-1}$;
    **if** $c(f_\theta, Q'(b, a_1, a_2)) > c(f_\theta, Q^*)$ **then**
      $\tilde{Q} \leftarrow Q'(b, a_1, a_2)$
      # update the best attacked problem
  Sort $Q'(\cdot, \cdot, \cdot)$ w.r.t. cost values by decreasing order;
  $Q_{1..B}^k \leftarrow Q'_{1..B}$; # select top-$B$ for next step
**Output:** Best attacked problem instance $\tilde{Q}$.

---

## 3 STUDIED CO TASKS AND RESULTS

We conduct experiments on four representative CO tasks: Directed Acyclic Graph Scheduling, Asymmetric Traveling Salesman Problem, Maximum Coverage and Maximum Coverage with Separate Coverage Constraint. Considering the page limitation, the details of the last task is postponed to Appendix C. We also test the robustness of representative solvers including heuristics (Helsgaun, 2017; Grandl et al., 2014), branch-and-bound solvers (Forrest et al., 2022; The SCIP Optimization Suite 8.0, 2021; Gurobi Optimization, 2020) and learning-based methods (Kwon et al., 2021). The detailed graph embedding methods for the four tasks is shown in Appendix D. In Appendix I, we provide training and evaluation hyperparameters of different solvers for fair time comparison and reproducibility. Experiments are run on our heterogeneous cluster with RTX 2080Ti and RTX 3090.

**Metrics.** The quantitative robustness has been defined in Def. 2, and here we propose a more intuitive metric of robustness in terms of ratios, which is also against the attacker's objective. Typically, we just adopt the relative change rate between the cost value after attacked and the cost of the incumbent optimal solution, i.e. the best solution we can find, so that the ratio is proportional to the attacker's objective. In the result tables, we report the raw cost value of different solvers under different attacks. Besides, we also report the robustness ratio for ROCO-RL, which is the best attacker in most cases.

### 3.1 TASK I: DIRECTED ACYCLIC GRAPH SCHEDULING

Many task scheduling systems formulate the job stages and their dependencies as a Directed Acyclic Graph (DAG) (Saha et al., 2015; Chambers et al., 2010), as shown in Fig. 4. The data center has limited computing resources to allocate the jobs with different resource requirements. These jobs can run in parallel if all their parent jobs have finished and the required resources are available. Our goal is to **minimize the makespan (i.e. finish all jobs as fast as possible.)**

**Solvers.** We choose three popular heuristic solvers as our attack targets. First, the Shortest Job First algorithm chooses the jobs greedily with minimum completion time. Second, the Critical Path algorithm finds the bottlenecks and finishes the jobs in the critical path sequence. Third, the Tetris

Table 2: Evaluation on DAG scheduling. Clean makespans are results on the clean test set. Stronger attackers result in larger attacked makespans. Random denotes the random attack baseline with the mean and std tested for 100 trials. ROCO-RA/OG/SA are tested for 10 trials to calculate the mean and std. ROCO-RL is reported with a single value since it is invariant to random seeds. The ratio in red is the robustness raio defined in metrics. A lower ratio denotes a more robust solver.

| Solver | Size: #job | Clean Makespan ($\times 10^5$) ↓ | Attack Method (Attacked Makespans ($\times 10^5$)) | | | | |
|---|---|---|---|---|---|---|---|
| | | | Random | ROCO-RA | ROCO-OG | ROCO-SA | ROCO-RL |
| Shortest Job First | 50 | 1.0461 | $1.0406 \pm 0.0035$ | $1.0574 \pm 0.0013$ | $1.0600 \pm 0.0019$ | $\mathbf{1.0622 \pm 0.0007}$ | 1.0608 ($r = 28.94\%$) |
| Critical Path | 50 | 0.8695 | $0.8828 \pm 0.0077$ | $0.9402 \pm 0.0038$ | $0.9480 \pm 0.0021$ | $\mathbf{0.9528 \pm 0.0013}$ | 0.9500 ($r = 15.47\%$) |
| Tetris | 50 | 0.8227 | $0.8582 \pm 0.0079$ | $0.8952 \pm 0.0049$ | $0.9218 \pm 0.0066$ | $0.9380 \pm 0.1153$ | $\mathbf{0.9397}$ ($r = 14.22\%$) |
| Shortest Job First | 100 | 1.9160 | $1.8832 \pm 0.0075$ | $1.9210 \pm 0.0006$ | $1.9239 \pm 0.0008$ | $1.9239 \pm 0.0008$ | $\mathbf{1.9263}$ ($r = 26.85\%$) |
| Critical Path | 100 | 1.6018 | $1.6279 \pm 0.0167$ | $1.7391 \pm 0.0045$ | $1.7456 \pm 0.0043$ | $1.7480 \pm 0.0003$ | $\mathbf{1.7498}$ ($r = 15.22\%$) |
| Tetris | 100 | 1.5186 | $1.5830 \pm 0.0196$ | $1.7201 \pm 0.0055$ | $1.7099 \pm 0.0111$ | $1.7418 \pm 0.0074$ | $\mathbf{1.7526}$ ($r = 15.41\%$) |
| Shortest Job First | 150 | 2.8578 | $2.8281 \pm 0.0074$ | $2.8818 \pm 0.0020$ | $2.8898 \pm 0.0023$ | $2.8950 \pm 0.0014$ | $\mathbf{2.8964}$ ($r = 28.91\%$) |
| Critical Path | 150 | 2.4398 | $2.4254 \pm 0.0203$ | $2.5698 \pm 0.0093$ | $2.5928 \pm 0.0090$ | $2.6020 \pm 0.0029$ | $\mathbf{2.6069}$ ($r = 16.02\%$) |
| Tetris | 150 | 2.2469 | $2.3581 \pm 0.0270$ | $2.4988 \pm 0.0191$ | $2.5039 \pm 0.0202$ | $\mathbf{2.5399 \pm 0.0058}$ | 2.5329 ($r = 12.73\%$) |

(Grandl et al., 2014) scheduling algorithm models the jobs as 2-dimension blocks in the Tetris games according to their finishing time and resource requirement.

**Attack model.** The edges in a DAG represent job dependencies, and removing edges will relax the constraints. After removing existing edges in a DAG, it is obvious that the new solution will be equal to or better than the original with fewer restrictions. Therefore, the attack model selectively removes existing edges.

**Dataset.** We use TPC-H dataset[1], composed of business-oriented queries and concurrent data modification. Many DAGs referring to computation jobs, have tens or hundreds of stages with different duration and numbers of parallel tasks.

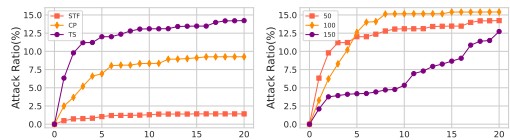

Figure 2: Left: RL Attack results w.r.t. attack budgets $K$ on TPC-H-50 dataset for solvers Shortest Job First(SJF), Critical Path(CP) and Tetris(TS). Attack ratio is the ratio of the increased makespan after the attack to the clean makespan. Right: RL Attack results w.r.t. attack budgets $K$ on TPC-H-50/100/150 datasets for the Tetris (TS) solver.

We gather the DAGs randomly and generate three different sizes of datasets, TPC-H-50, TPC-H-100, TPC-H-150, each with 50 training and 10 testing samples. The DAG nodes have two properties: execution time and resource requirement.

**Results and analysis.** Tbl. 2 reports the results of our four attack methods. In the designed attack methods, RL outperforms other learning-free methods in most cases, illustrating the correctness of our feature extraction techniques and training framework. It is worth noting that even the simplest random search (ROCO-RA) can cause a significant performance degradation to the CO solvers, showing their vulnerability and the effectiveness of the attack framework. Besides, all four attack methods perform better than the random attack baseline, which usually leads to a shorter makespan than the clean instances (unsuccessful attack), proving that our attack framework can find more hard instances than those because of the inherent instability of CO problems. Fig. 2 (left) shows the attack ratio for different solvers on TPC-H-50. In general, the attack ratio increases then flattens out w.r.t. attack budgets, corresponding with the fact that we can find more hard instances under a larger searching space (but harder to find new successfully attacked instances as we remove more constraints). Fig. 2 (right) demonstrates the effect of attack budgets on different sizes of datasets using the solver Tetris. The figure illustrates that attack ratios on smaller datasets tend to flatten first (larger datasets allow for more attack actions), which can guide us to select the right number of attack actions for different datasets. We also conduct experiments to verify the generalizability of ROCO-RL in Appendix L, which shows that our attacker can be trained efficiently on small problem instances, then be directly used on larger problems.

## 3.2 TASK II: ASYMMETRIC TRAVELING SALESMAN

The classic traveling salesman problem (TSP) is to **find the shortest cycle to travel across all the cities.** Here we tackle the extended version asymmetric TSP (ATSP) for generality.

**Solvers.** The robustness of four algorithms are evaluated: First, Nearest Neighbour greedily adds the nearest city to the tour. Second, Furthest Insertion finds the city with the furthest distance to existing cities in the tour and inserts it. Third, Lin-Kernighan Heuristic (LKH3) (Helsgaun, 2017) is

[1]http://tpc.org/tpch/default5.asp

Table 3: Evaluation of solvers for ATSP. Stronger attacker result in larger attacked tour lengths. MatNet (fixed) represents training MatNet with 10000 fixed ATSP instances.

| Solver | Problem Size: # city | Clean Tour $(\times 10^6)\downarrow$ | Attack Method (Attacked Tour Length $(\times 10^6)$) | | | | |
|---|---|---|---|---|---|---|---|
| | | | Random | ROCO-RA | ROCO-OG | ROCO-SA | ROCO-RL |
| Nearest Neighbour | 20 | 1.9354 | $1.9143 \pm 0.0259$ | $2.1307 \pm 0.0153$ | $2.1162 \pm 0.0323$ | $2.1344 \pm 0.0159$ | **2.1858**($r = 49.76\%$) |
| Furthest Insertion | 20 | 1.6092 | $1.5847 \pm 0.0142$ | $1.6953 \pm 0.0105$ | $1.6926 \pm 0.0117$ | $1.7183 \pm 0.0114$ | **1.7469** ($r = 19.69\%$) |
| LKH3 | 20 | 1.4595 | $1.4201 \pm 0.0089$ | $1.4599 \pm 0.0003$ | $1.4599 \pm 0.0004$ | $1.4610 \pm 0.0017$ | **1.4611** ($r = 0.11\%$) |
| MatNet | 20 | 1.4616 | $1.4237 \pm 0.0089$ | $1.4674 \pm 0.0012$ | $1.4683 \pm 0.0006$ | $1.4683 \pm 0.0009$ | **1.4708** ($r = 0.77\%$) |
| MatNet (fixed) | 20 | 1.4623 | $1.4227 \pm 0.0086$ | $1.4665 \pm 0.0013$ | $1.4687 \pm 0.0004$ | $1.4714 \pm 0.0013$ | **1.4737** ($r = 0.97\%$) |
| Nearest Neighbour | 50 | 2.2247 | $2.2221 \pm 0.0131$ | $2.3635 \pm 0.0082$ | $2.3809 \pm 0.0096$ | $2.4058 \pm 0.0151$ | **2.4530** ($r = 47.58\%$) |
| Furthest Insertion | 50 | 1.9772 | $1.9730 \pm 0.0091$ | $2.0593 \pm 0.0071$ | $2.0466 \pm 0.0125$ | $2.0632 \pm 0.0089$ | **2.1150** ($r = 27.25\%$) |
| LKH3 | 50 | 1.6621 | $1.6493 \pm 0.0042$ | $1.6653 \pm 0.0007$ | $1.6656 \pm 0.0007$ | **1.6682 $\pm$ 0.0010** | 1.6679 ($r = 0.35\%$) |
| MatNet | 50 | 1.6915 | $1.6786 \pm 0.0051$ | $1.7153 \pm 0.0012$ | $1.7204 \pm 0.0016$ | $1.7255 \pm 0.0012$ | **1.7279** ($r = 3.96\%$) |
| MatNet(fixed) | 50 | 1.6972 | $1.6894 \pm 0.0054$ | $1.7171 \pm 0.0021$ | $1.7205 \pm 0.0005$ | $1.7279 \pm 0.0007$ | **1.7349** ($r = 4.38\%$) |
| Nearest Neighbour | 100 | 2.1456 | $2.1423 \pm 0.0079$ | $2.2319 \pm 0.0099$ | $2.2213 \pm 0.0152$ | $2.2273 \pm 0.0112$ | **2.2533** ($r = 42.95\%$) |
| Furthest Insertion | 100 | 1.9209 | $1.9188 \pm 0.0061$ | $1.9762 \pm 0.0088$ | $1.9782 \pm 0.0111$ | $1.9853 \pm 0.0063$ | **2.0144** ($r = 27.79\%$) |
| LKH3 | 100 | 1.5763 | $1.5716 \pm 0.0016$ | $1.5826 \pm 0.0006$ | $1.5848 \pm 0.0005$ | $1.5856 \pm 0.0003$ | **1.5862** ($r = 0.62\%$) |
| MatNet | 100 | 1.6545 | $1.6492 \pm 0.0036$ | $1.6772 \pm 0.0012$ | $1.6815 \pm 0.0005$ | $1.6841 \pm 0.0007$ | **1.6873** ($r = 7.04\%$) |
| MatNet (fixed) | 100 | 1.6556 | $1.6498 \pm 0.0037$ | $1.6788 \pm 0.0008$ | $1.6823 \pm 0.0003$ | $1.6846 \pm 0.0015$ | **1.6919** ($r = 7.33\%$) |

the traditional SOTA TSP solver. Finally, Matrix Encoding Networks (MatNet) (Kwon et al., 2021) claims as a SOTA learning-based solver for ATSP and flexible flow shop (FFSP).

**Attack model.** The attack is to choose an edge and half its value, by which we get a no-worse theoretical optimum since the length of any path will maintain the same or decrease. To reduce the action space, we will not select the edges in the current path predicted by the solver at the last time step. An attack example is shown in Fig. 5 in Appendix E.

**Dataset.** The dataset comes from (Kwon et al., 2021), consisting of 'tmat' class ATSP instances which have the triangle inequality and are widely studied by the operation research community (Cirasella et al., 2001). We solve the ATSP of three sizes, 20, 50 and 100 cities. The distance matrix is fully connected and asymmetric, and each dataset consists of 50 training samples and 20 testing samples.

**Results and analysis.** Tbl. 3 reports the attack results of four target solvers. where the RL based attack outperforms other methods in most cases. Note that LKH3 is the most robust among all compared solvers. We attribute the phenomenon to two aspcets of LKH3: *Different initial states-* LKH3 will take 4 runs with diffenent initial states in our setting; *Step by step improvement-* Specifically, LKH3 uses k-opt (Lin, 1965) to iteratively reorder and optimize the result tour sequence. The introduction of randomization and optimization should contribute to the strong robustness of LKH3. Inspired by this idea, we conduct experiments to improve the robustness of Nearest Neighbor and Furtherest Insertion in Tbl. 9 and Tbl. 10. MatNet generates different random problem instances for training in each iteration. The comparison between "MatNet" and "MatNet (fixed)" proves that the huge amount of i.i.d. training instances promotes both the performance and the robustness of the model. As Yehuda et al. (2020) points out, learning-based solvers for CO cannot be trained sufficiently due to the theoretical limitation of data generation. Thus, the training paradigm with large unlabeled data can be a good choice for learning-based CO solvers. Besides, feeding hard samples discovered by ROCO to "MatNet (fixed)" to conduct adversarial training may also promote robustness (see Tbl. 11).

### 3.3 TASK III: MAXIMUM COVERAGE

The maximum coverage (MC) problem is widely studied in approximation algorithms. Specifically, as input we will be given several sets $\mathcal{S} = \{S_1, S_2, ..., S_m\}$ and a number $k$. The sets consist of weighted elements $e \in \mathcal{E}$ and may have overlap with each other. Our goal is to **select at most $k$ of these sets such that the maximum weight of elements are covered.** The above problem can be formulated as follows, where $w(\cdot) : \mathcal{E} \to \mathbb{R}$ denotes the weight of a certain element.

$$\max_{\mathcal{S}'} \sum_{e \in \mathcal{E}} w(e) \times \mathbb{I}(e \in \bigcup_{S_i \in \mathcal{S}'} S_i) \quad s.t. \quad \mathcal{S}' \subseteq \mathcal{S}, |\mathcal{S}'| \leq k \tag{9}$$

**Solvers.** It has been shown that the greedy algorithm choosing a set that contains the largest weight of uncovered elements achieves an approximation ratio of $(1 - \frac{1}{e})$ (Hochbaum, 1996). which has also been proved to be the best-possible polynomial-time approximation algorithm for maximum coverage unless P=NP (Feige, 1998). So we choose it to serve as our first attack target. Besides, the maximum coverage problem is quite suitable to be formulated as an integer linear program (ILP). Using Google ORTools API[2], we transform Eq. 9 into the ILP form and attack three other general-purpose solvers:

---

[2]https://developers.google.com/optimization

Table 4: Evaluation for MC. The time limit is in brackets for B&B solvers, which is set long enough to output a feasible solution in the clean dataset. Stronger attackers result in smaller attacked weights.

| Solver | Problem Size: #set-#element | Clean Weight ($\times 10^4$) ↑ | Attack Method (Attacked Weight ($\times 10^4$)) | | | | |
|---|---|---|---|---|---|---|---|
| | | | Random | ROCO-RA | ROCO-OG | ROCO-SA | ROCO-RL |
| Greedy | 100-200 | 0.7349 | $0.7353 \pm 0.0007$ | $0.7252 \pm 0.0012$ | $0.7255 \pm 0.0013$ | $0.7253 \pm 0.0007$ | **0.7239** ($r = 3.36\%$) |
| CBC(0.5s) | 100-200 | 0.7115 | $0.7020 \pm 0.0122$ | $0.6161 \pm 0.0206$ | $0.5448 \pm 0.0255$ | $0.5423 \pm 0.0315$ | **0.5029** ($r = 32.87\%$) |
| SCIP(0.5s) | 100-200 | 0.7184 | $0.7116 \pm 0.0053$ | $0.6850 \pm 0.0013$ | $0.6808 \pm 0.0025$ | $0.6803 \pm 0.0015$ | **0.6788** ($r = 9.38\%$) |
| Gurobi(0.5s) | 100-200 | 0.7491 | $0.7495 \pm 0.0004$ | $0.7460 \pm 0.0004$ | $0.7441 \pm 0.0006$ | $0.7426 \pm 0.0008$ | **0.7409** ($r = 1.09\%$) |
| Greedy | 150-300 | 1.1328 | $1.1330 \pm 0.0011$ | $1.1207 \pm 0.0019$ | $1.1183 \pm 0.0021$ | $1.1201 \pm 0.0017$ | **1.1182** ($r = 2.64\%$) |
| CBC(1s) | 150-300 | 1.0510 | $1.0181 \pm 0.0192$ | $0.8975 \pm 0.0386$ | $0.8054 \pm 0.0487$ | $0.7959 \pm 0.0482$ | **0.7527** ($r = 34.46\%$) |
| SCIP(1s) | 150-300 | 1.0847 | $1.0867 \pm 0.0027$ | $1.0544 \pm 0.0017$ | $1.0467 \pm 0.0016$ | $1.0486 \pm 0.0012$ | **1.0443** ($r = 9.07\%$) |
| Gurobi(1s) | 150-300 | 1.1485 | $1.1486 \pm 0.0009$ | $1.1391 \pm 0.0008$ | $1.1358 \pm 0.0011$ | $1.1332 \pm 0.0005$ | **1.1314** ($r = 1.49\%$) |
| Greedy | 200-400 | 1.4659 | $1.4665 \pm 0.0013$ | $1.4537 \pm 0.0019$ | $1.4533 \pm 0.0019$ | $1.4552 \pm 0.0013$ | **1.4466** ($r = 2.84\%$) |
| CBC(1.5s) | 200-400 | 1.4248 | $1.4088 \pm 0.0219$ | $1.2053 \pm 0.0475$ | $1.0966 \pm 0.0575$ | $1.0866 \pm 0.0705$ | **1.0418** ($r = 30.03\%$) |
| SCIP(1.5s) | 200-400 | 1.3994 | $1.3952 \pm 0.0049$ | $1.3636 \pm 0.0013$ | $1.3608 \pm 0.0017$ | $1.3629 \pm 0.0018$ | **1.3536** ($r = 9.09\%$) |
| Gurobi(1.5s) | 200-400 | 1.4889 | $1.4879 \pm 0.0013$ | $1.4754 \pm 0.0022$ | $1.4721 \pm 0.0018$ | $1.4697 \pm 0.0003$ | **1.4684** ($r = 1.38\%$) |

Gurobi (the SOTA commercial solver) (Gurobi Optimization, 2020), SCIP (The SCIP Optimization Suite 8.0, 2021), and CBC (an open-sourced MILP solver written in C++) (Forrest et al., 2022), to find their vulnerabilities. These solvers are set the same appropriate time limit.

**Calibrated time.** In line with Nair et al. (2020), we use calibrated time to measure running time of task solving to reduce the interference of backend running scripts and the instability of the server. The main idea is to solve a small calibration MIP continuously and use the average running time to measure the speed of the machine when solving different instances. Details are given in Appendix G.

**Attack model.** The MC problem can achieve a no-worse optimum if we add new elements into certain sets, as we can cover more elements while not exceeding the threshold $k$. MC can be treated as a bipartite graph, where edges only exist between a set node and an element node inside the set. So our attack model is to add edges between set nodes and elements nodes, leading to a theoretically better optimum but can mislead the solvers. To reduce the action space, we only add edges for the unchosen sets, since adding edges for selected sets will not affect the solver's solution.

**Dataset.** For the MC problem, the distribution follows ORLIB[3]. The dataset consists of three set-element pairs 100-200, 150-300, and 200-400, each with 50 training and 20 testing samples.

**Results and analysis.** Tbl. 4 records the attack results of four target solvers. The RL based attack outperforms other methods across all cases in this scenario, showing the promising power of our feature extraction method and policy gradient model. Besides, the attackers have even found some instances for CBC where it cannot figure out feasible solutions under the given time limit, strongly demonstrating the vulnerability of the solvers. This effect also appears in the more challenging task Maximum Coverage with Separate Coverage Constraint MCSCC for Gurobi in Appendix C. Typically, our attack model to MC problems is equivalent to modifying the objective coefficients of the original ILP model of MC problem, which might incur numerical issues for ILP solver such as unstable matrix factorization. Since we can access the source code of CBC solver and SCIP solver respectively, we find that the SCIP solver is equipped with more modules improving numerical stability of solving process, such as scaling, presolving and primal heuristics, etc. Besides, SCIP also benefits from implementing more classes of branching policy and cutting planes that aim to improve the dual bound, than CBC. The experimental results are in line with the expectation for the tested ILP solvers, where CBC solver performs worse than SCIP solver in terms of both solution quality and robustness. Compared to the open-sourced solvers, the commercial solver Gurobi performs undoubtedly better in the perspective of both solution quality and robustness. One step further, we conduct experiments on tuning the hyperparameters of Gurobi to improve its robustness in Tbl. 12.

## 4 CONCLUSION

We propose Robust Combinatorial Optimization (ROCO), the first general framework to evaluate the robustness of combinatorial solvers on graphs without requiring optimal solutions or the differentiable property. Alleviating these two requirements makes it generalized and flexible, helping us conduct an extensive evaluation of the robustness of 14 unique combinations of different solvers and problems, showing the effectiveness and potential of our methods. Moreover, we also provide insights to the robustness of solvers and conduct various experiments to improve their robustness, which we believe can encourage future works to develop more robust CO solvers.

---

[3] http://people.brunel.ac.uk/~mastjjb/jeb/info.html

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

## A    RELATED WORKS

**Combinatorial optimization solvers.** Traditional CO algorithms include but not limited to greedy algorithms, heuristic algorithms like simulated annealing (SA) (Van Laarhoven & Aarts, 1987), Lin–Kernighan–Helsgaun (LKH3) (Helsgaun, 2017), and branch-and-bound solvers like CBC (Forrest et al., 2022), SCIP (The SCIP Optimization Suite 8.0, 2021), and Gurobi (Gurobi Optimization, 2020). Besides, learning-based methods have also been proposed to tackle these problems. A mainstream approach using deep learning is to predict the solution end-to-end, such as the supervised model Pointer Networks (Vinyals et al., 2015), reinforcement learning models S2V-DQN (Khalil et al., 2017) and MatNet (Kwon et al., 2021). Though these methods did perform well on different types of CO problems, they are not that robust and universal, as discussed in Bengio et al. (2021), the solvers may get stuck around poor solutions in many cases. The sensitivity of CO algorithms is theoretically characterized in Varma & Yoshida (2021), however the metric in Varma & Yoshida (2021) requires optimal solutions which are usually unavailable in practice concerning the NP-hard challenge in CO.

**Adversarial attack and defense for neural networks.** Since the seminal study Szegedy et al. (2014) shows that small input perturbations can change model predictions, many adversarial attack methods have been devised to construct such attacks. In general, adversarial attacks can be roughly divided into two categories: white-box attacks with access to the model gradients, e.g. Goodfellow et al. (2015); Madry et al. (2018); Carlini & Wagner (2017), and black-box attacks, with only access to the model predictions, e.g. Ilyas et al. (2018); Narodytska & Kasiviswanathan (2016). Besides image and text adversarial attacks (Jia & Liang, 2017), given the importance of graph-related applications and the successful applications of graph neural networks (GNN) (Scarselli et al., 2008), more attention is recently paid to the robustness of GNNs (Dai et al., 2018). Orthogonal to designing more effective attacks, there is an interesting line of works that seek to inspect the nature of attacks, such as through the frequency perspective (Chen et al., 2022b; Lin et al., 2022; Yash Sharma, 2019).

In the mean time, many defense strategies like adversarial training Ganin et al. (2016); Tramèr et al. (2020); Ren et al. (2022b) have also been proposed to counter this series of attack methods. Besides, as graph matching (GM) is a classic CO problem, Ren et al. (2022a) shows how to fulfill attack and defense on GM, which reveals the vulnerability of deep GM by perturbing visual pixels and graph structure iteratively and improves the robustness of deep GM by separating appearance-similar nodes to be distributed more broadly.

## B    TRADITIONAL ATTACK ALGORITHM

As an example of traditional attack algorithm, we list the pseudo code of SA in Algorithm 2.

---

**Algorithm 2: Simulated Annealing (SA) Attack**

**Input:** Input problem $Q$; solver $f_\theta$; max number of actions $K$; action sample number $M$;
        Temperature decay $\Delta T$; coefficient $\beta$; bias $eps$.
$Q^0 \leftarrow Q; Q^* \leftarrow Q^0; T \leftarrow 1;$ # initial temperature
**for** $k \leftarrow 1..K$ **do**
    flag = False; # if action is available
    **for** $i \leftarrow 1..M$ **do**
       Random sample an edge $(x, y)$ in edge candidates of $Q^{k-1}$;
       $\tilde{Q} \leftarrow$ add/delete the edge $(x, y)$ in $Q^{k-1}$; # new state by tentative action
       $P = \exp(\frac{\beta(c(f_\theta, \tilde{Q}) - c(f_\theta, Q^{k-1}) + eps)}{T})$; # action acceptance probability
       **if** $\text{Random}(0, 1) \leq P$ **then**
          flag = True; $Q^k \leftarrow \tilde{Q}; Q^* \leftarrow Q^k$
          **break**;
    **if** flag = False **then**
       **break**;
    $T = T \cdot \Delta T;$
**Output:** Problem instance $\mathcal{G}^*$.

---

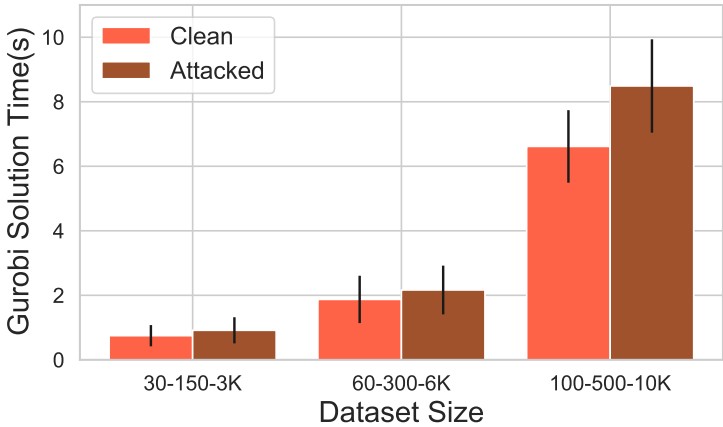

Figure 3: Gurobi's average time cost (no time limit) in solving clean or attacked MCSCC problems. Experiments are run on 3 datasets (20 instances) of different sizes.

## C  EXPERIMENTS-TASK IV: MAXIMUM COVERAGE WITH SEPARATE COVERAGE CONSTRAINTS

We also study a more complicated version of the former MC problem: the maximum coverage problem with separate coverage constraint (MCSCC), which is NP-hard as proved in Appendix F.2. Specifically, the elements $\mathcal{E}$ can be classified into the black ones $\mathcal{B}$ and the white ones $\mathcal{W}$ (i.e. $\mathcal{E} = \mathcal{W} \cup \mathcal{B}, \mathcal{W} \cap \mathcal{B} = \emptyset$). And our goal is to **select a series of sets $\mathcal{S}' \subseteq \mathcal{S}$ to maximize the coverage of black element weights while covering no more than $k$ white elements.** The problem can be represented by a bipartite graph, where edges only exist between a set node and an element node covered by the set. The problem can be formulated as:

$$\max_{\mathcal{S}'} \sum_{b \in \mathcal{B}} w(b) \times \mathbb{I}(b \in \bigcup_{S_i \in \mathcal{S}'} C^+(S_i))$$

$$s.t. \quad \mathcal{S}' \subseteq \mathcal{S}, \; | \bigcup_{S_i \in \mathcal{S}'} C^-(S_i)| \le k \tag{10}$$

where $w(\cdot)$ denotes the weight of a certain element, $C(\cdot)$ denotes the set of elements covered by a set, and $C^+(\cdot)$ and $C^-(\cdot)$ denotes the set of elements in $C(\cdot)$ with black and white labels, respectively.

**Solvers.** As an enhanced version of MC, the MCSCC problem is very challenging and here we propose three different solvers as the target for attacking. First, the trivial Local algorithm iterates over the sets sequentially, adding any sets that will not exceed the threshold $k$. Second, we adopt a more intelligent Greedy Average algorithm that always chooses the most cost-effective (the ratio of the increase of black element weights to the increase of a number of covered white elements) set at each step until the threshold $k$ is exceeded. Third, we formulate the problem into the standard ILP form (details in Appendix F.3) and solve it by Gurobi (Gurobi Optimization, 2020).

**Attack model.** Our attack model chooses to add non-existing black edges that connect sets to black elements, which can lead to a theoretically better optimum since we can possibly cover more black elements while not exceeding the threshold $k$. Further, in order to reduce the action space, we only select the unchosen sets, otherwise it will be useless since adding edges for selected sets will not affect a solver's output solution. A successful attack example is shown in Fig. 6 in Appendix E.

**Dataset.** There is a lack of large-scale real-world datasets for MCSCC. Thus, we keep the data distribution similar to the MC problem and randomly generate the dataset (Kwon et al., 2021) for training. Specifically, the dataset consists of three configs of set-black element-white element: 30-150-3K, 60-300-6K and 100-500-10K, each with 50 training samples and 10 testing samples. More details about the dataset are presented in Appendix H.

**Results and analysis.** Tbl. 5 shows the attack results on our simulated dataset. Both traditional and RL approaches achieve significant attack effects, while RL outperforms the others in most cases

Table 5: Evaluation of solvers for MCSCC (notations are kept in line with MC in Table 4). Since MCSCC incorporates more constraints and is more challenging, our ROCO attackers show a more significant effect compared to MC, suggesting that solvers may be more vulnerable on CO problems that are inherently more challenging (e.g. with more complicated constraints). The greedy average algorithm is the most robust probably due to its worst-case guarantee known as submodularity (Hochbaum, 1996).

| Solver | Problem Size: #set-#b elem.-#w elem. | Clean Weight ↑ | Attack Method (Attacked Weight) | | | | |
|---|---|---|---|---|---|---|---|
| | | | Random | ROCO-RA | ROCO-OG | ROCO-SA | ROCO-RL |
| Local Search | 30-150-3K | 9.5713 | $9.5474 \pm 0.0325$ | $9.4966 \pm 0.0057$ | $9.4976 \pm 0.0105$ | $9.4899 \pm 0.0029$ | **9.4861**($r = 49.79\%$) |
| Greedy Average | 30-150-3K | 18.0038 | $18.2541 \pm 0.1710$ | $17.5141 \pm 0.0288$ | $17.4331 \pm 0.0414$ | $17.5177 \pm 0.0396$ | **17.1414** ($r = 9.28\%$) |
| Gurobi(1s) | 30-150-3K | 18.8934 | $19.2297 \pm 0.1795$ | $16.9266 \pm 0.2135$ | $15.4132 \pm 0.3552$ | $15.3055 \pm 0.3684$ | **14.2740** ($r = 24.45\%$) |
| Local Search | 60-300-6K | 24.9913 | $25.0413 \pm 0.0625$ | $24.8738 \pm 0.0100$ | **24.7914 ± 0.035** | $24.8189 \pm 0.0375$ | 24.8014 ($r = 42.54\%$) |
| Greedy Average | 60-300-6K | 43.1625 | $43.1539 \pm 0.0475$ | $42.7697 \pm 0.0388$ | $42.7611 \pm 0.0475$ | $42.7222 \pm 0.0388$ | **42.1741** ($r = 2.29\%$) |
| Gurobi(2s) | 60-300-6K | 41.1828 | $41.8005 \pm 0.4407$ | $38.2382 \pm 0.3459$ | $37.3322 \pm 0.4201$ | $38.2918 \pm 0.3665$ | **36.0432** ($r = 16.49\%$) |
| Local Search | 100-500-10K | 22.9359 | $22.9634 \pm 0.0482$ | $22.7616 \pm 0.0206$ | $22.6538 \pm 0.0183$ | $22.7455 \pm 0.0206$ | **22.5804** ($r = 56.06\%$) |
| Greedy Average | 100-500-10K | 51.3905 | $51.4008 \pm 0.0308$ | $50.7481 \pm 0.0719$ | **50.5169 ± 0.1747** | $50.6865 \pm 0.0411$ | 50.5631 ($r = 1.61\%$) |
| Gurobi(5s) | 100-500-10K | 49.3296 | $52.8192 \pm 0.3803$ | $48.1375 \pm 0.3597$ | $47.4386 \pm 0.4933$ | $49.2013 \pm 0.2467$ | **43.4866** ($r = 15.38\%$) |

(especially for Gurobi). We can see from Tbl. 5 that the degradation of Gurobi after attack is much more obvious than that in MC, possibly because the computation progress of MCSCC can include more vulnerable steps than MC. Fig. 3 shows the average time cost (no time limit) for Gurobi in solving MCSCC problems. The figure illustrates that solving time on the attacked instances is longer than those on the clean ones by a large margin, proving the promising power of our attack framework.

## D  GRAPH EMBEDDING FOR SPECIFIC TASKS

We discuss specific graph embedding metrics for different kinds of CO problems for reproducibility.

### D.1  TASK I: DIRECTED ACYCLIC GRAPH SCHEDULING

Since the task is a directed acyclic graph, we use GCN to encode the state in the original graph and its reverse graph with inversely directed edges separately. Then we concatenate the two node embeddings and use an attention pooling layer to extract the graph-level embedding for Eq. 6:

$$\mathbf{n} = [\text{GCN}_1(\mathcal{G})||\text{GCN}_2(\text{reverse}(\mathcal{G}))], \ \mathbf{g} = \text{AttPool}(\mathbf{n}). \tag{11}$$

### D.2  TASK II: ASYMMETRIC TRAVELING SALESMAN PROBLEM

Since the graph is fully connected, we use GCN to encode the state in the graph. Then we use an attention pooling layer to extract the graph-level embedding. Eq. 6 becomes:

$$\mathbf{n} = [\text{GCN}(\mathcal{G})], \ \mathbf{g} = \text{AttPool}(\mathbf{n}). \tag{12}$$

### D.3  TASK III: MAXIMUM COVERAGE

For the RL attack method, different from DAG and ATSP, MC has a unique bipartite graph structure. Therefore, we resort to SAGEConv, which can handle bipartite data, for graph feature extraction. As input, we classify the nodes into two classes (subsets $I_s$ and elements $I_e$) and associate them with two dimension one-hot tensors. Besides, we add one more dimension for element nodes, which records their weights. Eq. 6 becomes:

$$\mathbf{n}_e = \text{SAGEConv}_1(I_s, I_e), \ \mathbf{n}_s = \text{SAGEConv}_2(I_e, I_s)$$
$$\mathbf{g}_e = \text{AttPool}_1(\mathbf{n}_e), \ \mathbf{g}_s = \text{AttPool}_2(\mathbf{n}_s) \tag{13}$$

### D.4  TASK IV: MAXIMUM COVERAGE WITH SEPARATE COVERAGE CONSTRAINTS

The graph embedding mechanism for MCSCC is exactly the same as MC since they can both be represented by a bipartite graph except that we add one more one-hot dimension to distinguish between black and white elements.

# E    SUCCESSFUL ATTACK EXAMPLES

## E.1    TASK I: DIRECTED ACYCLIC GRAPH SCHEDULING

A successful attack example is shown in Fig. 4.

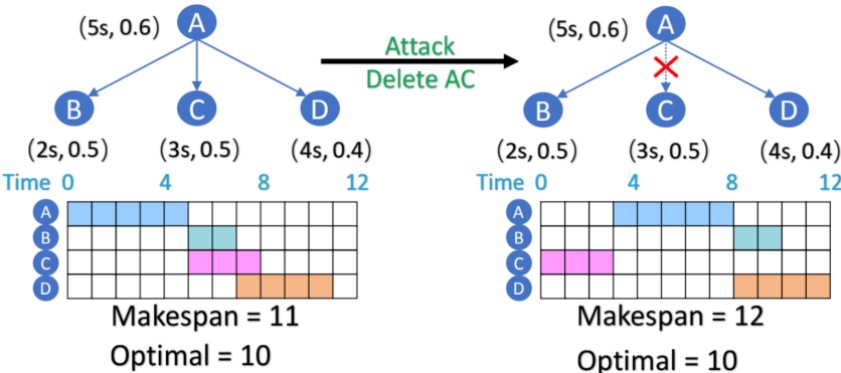

Figure 4: DAG attack on Shortest Job First algorithm. The edges show sequential dependencies. The $(x, y)$ tuple of each node means running time $(x)$ and resource occupancy rate $(y)$.

## E.2    TASK II: ASYMMETRIC TRAVELING SALESMAN PROBLEM

A successful attack example is shown in Fig. 5.

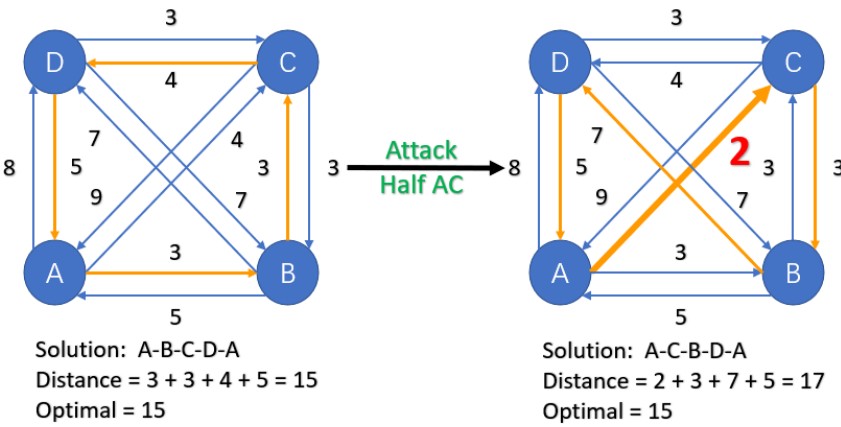

Figure 5: ATSP attack on Nearest Neighbour algorithm. The directed edge represents the distance between two nodes. The attack action on edge AC will cause 2 further distance.

## E.3    TASK III: MAXIMUM COVERAGE AND TASK IV: MAXIMUM COVERAGE WITH SEPARATE COVERAGE CONSTRAINTS

Successful attack examples are shown in Fig. 6.

# F    FORMULAS AND PROOFS

## F.1    NO-WORSE OPTIMAL SOLUTION GUARANTEE

We will prove that the attack constrains can guarantee no-worse optimal solution.

**Proposition 1.** *Given the clean CO problem $Q$ and the new instance $\tilde{Q}$ obtained by the attacker.*

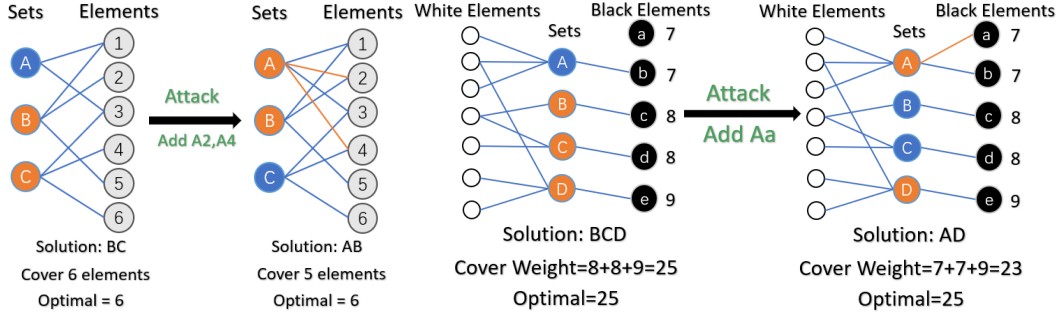

Figure 6: **Left**: MC attack on Greedy algorithm. The threshold $k$ of the sets is 2 and all the elements have unit weight. Attacks on edges A2 and A4 will cause a lower covered weight. **Right**: MCSCC attack on Greedy Average algorithm. The threshold $k$ of the white elements is 5 and the value near the black element represents its weight. Attack on edge Aa causes a lower covered black element weight.

1. *If the attacker removes some of the constraints or makes constraints easier to reach, which is defined as the first strategy **loosening the constraints**, the optimal solution will be no-worse.*

2. *If the attacker decreases the value of the objective function obtained from some feasible solutions, which is defined as the second strategy **lowering the cost**, the optimal solution will be no-worse.*

*Proof.* We define $\mathbf{x}^*$ and $\tilde{\mathbf{x}}^*$ as the optimal solution, $X$ and $\tilde{X}$ as the feasible solution set in $Q$ and $\tilde{Q}$. According to the formulation in Eq.1, we have

$$\min_{\mathbf{x}} c(\mathbf{x}, Q) \ \ s.t. \ \ h_i(\mathbf{x}, Q) \leq 0, \ i = 1, ..., I \tag{14}$$

$$\min_{\tilde{\mathbf{x}}} c(\tilde{\mathbf{x}}, \tilde{Q}) \ \ s.t. \ \ \tilde{h}_i(\tilde{\mathbf{x}}, \tilde{Q}) \leq 0, \ i = 1, ..., \tilde{I} \tag{15}$$

1. The attacker modifies the constraints $h_i$ to $\tilde{h}_i$ in three ways: 1) Stay the same. For each same solution $\mathbf{x}$, $\tilde{h}_i(\mathbf{x}, \tilde{Q}) = h_i(\mathbf{x}, Q)$. 2) Remove the constraint. The constraint $h_i(\mathbf{x}, Q)$ is no longer in $\tilde{Q}$. 3) Acquire lower value. For each same solution $\mathbf{x}$, $\tilde{h}_i(\mathbf{x}, \tilde{Q}) \leq h_i(\mathbf{x}, Q)$. Hence, the less than zero inequality is easier to achieve. Because if we have $h_i(\mathbf{x}, Q) \leq 0$, there must be $\tilde{h}_i(\mathbf{x}, \tilde{Q}) \leq 0$.

   Three parts of the constraints ensure each feasible solution $x \in X$ satisfies the constraints in $Q$ still satisfies the constraints in $\tilde{Q}$. Therefore, we have $X \subseteq \tilde{X}$. Since the cost function does not change and $\mathbf{x}^* \in \tilde{X}$, the optimal solution in new instance $\tilde{Q}$ can be at least the same. Meanwhile, The new feasible solution set $\tilde{X} - X$ may provide a better solution. In conclusion, the optimal solution will be no-worse.

2. The attacker decreases the cost function in the new instance $\tilde{Q}$ where $c(\mathbf{x}, \tilde{Q}) \leq c(\mathbf{x}, Q)$ for each instance $\mathbf{x}$. The constraints stay the same so that $X = \tilde{X}$. For $\mathbf{x}^*$, there is $c(\mathbf{x}^*, \tilde{Q}) \leq c(\mathbf{x}^*, Q)$ and $\mathbf{x}^* \in \tilde{X}$ which means the optimal solution in $\tilde{Q}$ will be no-worse.

□

### F.2 MCSCC NP-HARD PROVEMENT

We prove that the decision problem of **MCSCC** is NP-complete, thus the optimization problem of **MCSCC** in our paper is NP-hard. For the sake of our proof, here we redefine **MCSCC** and give definition of a traditional NP-Complete problem: **Set Covering**.

**The decision problem of MCSCC.** Given a series of sets $\mathcal{R} = \{R_1, R_2, ..., R_n\}$ along with an element set consisted of white and black elements $\mathcal{E} = \mathcal{W} \cup \mathcal{B}$. The sets cover certain elements (white $C^-(R_i)$ or black $C^+(R_i)$) which have their weights $w(e)$. Does there exist a collection of these sets $\mathcal{R}' \subseteq \mathcal{R}$ to cover $\geq M$ black elements weight while influence no more than $k$ white elements?

**Set Covering.** Given a set $U$ of elements, a collection $S_1, S_2, \ldots, S_n$ of subsets of $U$, and integer $k$, does there exist a collection of $\leq k$ of these sets whose union equals to $U$?

First, we need to show that **MCSCC** is NP. Given a collection of selected sets $\mathcal{R}' = \{R_1, R_2, \ldots, R_m\}$, we can simply traverse the set, recording the covered black and white elements. Then, we can assert whether the covered black element number is no more than $k$ and fraudulent monetary value is no less than $m$.

Since the certification process can be done in $O(n^2)$, we can tell that **MCSCC** is NP. Then, for NP-hardness, we can get it reduced from **Set Covering**.

Suppose we have a **Set Covering** instance, we construct an equivalent **MCSCC** problem as follows:

- Create $|U|$ black elements with weight 1, corresponding to elements in **Set Covering**.
- Create $n$ sets, set $C^+(R_i) = S_i$.
- Connect each set to a different white element of weight 1.
- Set the white element threshold $k_w$ equal to the set number threshold $k_s$.
- Set black element weight target $M = |U|$.

Suppose we find a collection of sets which meet the conditions of **MCSCC**, then we select subsets in **Set Covering** $S_i$ iff we select $R_i$. The total number of subsets is no more than $k_w(k_s)$ since the influenced white element number is equal to $|\mathcal{R}'|$. The subsets also cover $U$ since the covered black element weight (each with amount 1) is no less than $|U|$. Similarly, we can prove that we can find a suitable subset collection for **MCSCC** if we have found a collection of subsets that meet the conditions of **Set Covering**. So we can induce that **Set Covering** $\leq_p$ **MCSCC**.

Thus, we have proved that the decision problem of **MCSCC** is NP-Complete.

### F.3 MCSCC ILP FORMULATION

As discussed in the main text, $\mathcal{B}$ denotes the set of black elements, $\mathcal{W}$ represents the set of white elements, $\mathcal{S}$ refers to the collection of sets and $\mathcal{E}$ denotes the element set. Using notations above, we can translate Eq. 10 into standard ILP form as follows:

$$\max \sum_{i \in \mathcal{B}} Y[i] \times W[i], \quad s.t. \sum_{i \in \mathcal{W}} Y[i] \leq k$$

$$\text{for } i = 1 \ldots |\mathcal{E}|, \quad (Y[i] - 0.5)(0.5 - \sum_{j=1}^{|\mathcal{R}|} X[j] \times \mathbb{I}(i \text{ in } E[j])) \leq 0 \tag{16}$$

where $X[j] \in \{0, 1\}$ denotes set $S_j$ is chosen (1) or not (0), $Y[i] \in \{0, 1\}$ shows whether element $i$ has been covered by the chosen sets. Besides, $W[i] \in \mathbb{R}$ records the weight of the elements while $E[j] \subseteq \mathcal{E}$ is the corresponding elements of set $S_j$. The third equation ensures the element binary variable $Y[i]$ to be 1 iff it has been covered by a certain set (if $\exists X[j] = 1$ and element $i \in E[j]$, then the formula in the second bracket is negative, ensuring $Y[i]$ to be 1; else if element $i$ is not covered by any chosen sets, then the second formula is positive and $Y[i]$ must be 0).

## G CALIBRATED TIME

Considering the instability of the server and the interference of other running programs, calibrated time is designed to evaluate the server speed within a small period of time. Specifically, we define the speed as the reciprocal of the Wall clock time to solve a given calibration MIP. The calibration MIP is generated in the stable condition and solved $K$ times to record the basic speed $\text{speed}_{base}$. When evaluating the other instances, we will first evaluate the calibration MIP $K$ times to calculate the current speed $\text{speed}_{now}$. $K$ is the number of samples, which is set as 20 in our experiments.

To fairly measure the solvers' performance under different computational resources, take MC problems as an example, a given calibration MIP is also a specified MC problem to design time limits for different server speeds. Specifically, the new time limit for the solver will be set as:

$$\text{New Time Limit} = \frac{\text{speed}_{base}}{\text{speed}_{now}} \times \text{Uniform Time Limit} \tag{17}$$

## H  MCSCC DATASET

Besides the distribution rules of MC, we generate the MCSCC dataset mainly by two rules: 1) The black element weights are uniformly distributed in the range $[0, 1]$. 2) The number of black elements is 5% of white elements. 3) The covered black element weights and wight element numbers of sets follow a Gaussian distribution with a large standard deviation, to enlarge the differences between the sets.

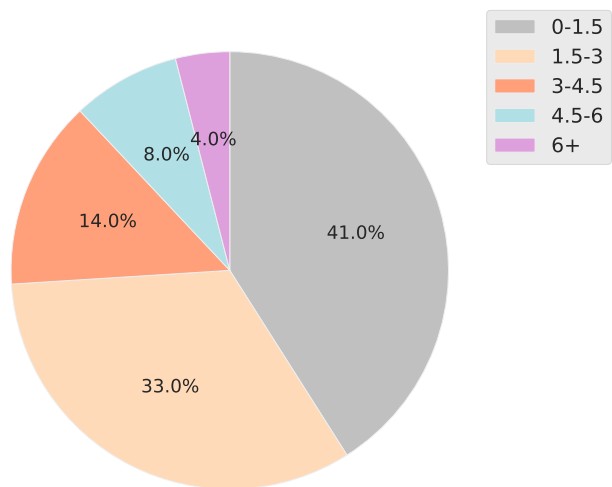

Figure 7: The distribution of black element weights covered by different sets. The $x - y$ pair in legends means the set can cover weights in range $[x, y]$.

Fig. 7 shows the black element weights covered by different sets in a 100-500-10K problem instance. As we can see, few well-designed sets can cover most weights while the others can be regarded as complementary to these sets. This design enlarges the difference between the different quality of solutions, therefore promoting the potential to be attacked.

## I  EXPERIMENT SETUPS

**Experiment environments.** DAG and ATSP experiments are run on a GeForce RTX 2080Ti while MC and MCSCC experiments are run on a GeForce RTX 3090 (20GB). We implement our models with Python 3.7, PyTorch 1.9.0 and PyTorch Geometric 1.7.2.

**RL settings.** Tbl. 6 records the hyperparameters for RL during the training process. Trust region clip factor is a parameter in PPO agent to avoid model collapse. We also adopt some common policy-gradient training tricks like reward normalization and entropy regularization during training processes.

**Attacker hyperparameters.** For fair comparison of different attackers and the consideration of RL inference time, the hyperparameters are set to ensure similar evaluation time across different attack methods. According to the time complexity we calculate in Tbl. 7, here we specify the following parameters: number of iterations $N$, beam search size $B$ and number of different actions $M$ in each iteration.

- **DAG**: ROCO-RA $N = 30$; ROCO-OG $B = 3$, $M = 9$; ROCO-SA $N = 5$, $M = 6$; ROCO-RL $B = 3$.
- **ATSP**: ROCO-RA $N = 130$; ROCO-OG $B = 5$, $M = 25$; ROCO-SA $N = 13$, $M = 10$; ROCO-RL $B = 5$.
- **MC**: ROCO-RA $N = 220$; ROCO-OG $B = 6$, $M = 36$; ROCO-SA $N = 22$, $M = 10$; ROCO-RL $B = 6$.

Table 6: RL parameter configuration in tasks DAG, ATSP, MC and MCSCC.

| Parameters | DAG | ATSP | MC & MCSCC |
|---|---|---|---|
| Actions# | 20 | 10 | 10 |
| Reward discount factor | 0.95 | 0.95 | 0.95 |
| Trust region clip factor | 0.1 | 0.1 | 0.1 |
| GNN type | GCN | GCN | SAGEConv |
| GNN layers# | 5 | 3 | 3 |
| Learning rate | 1e-4 | 1e-3 | 1e-3 |
| Node feature dimensions# | 64 | 20 | 16 |

Table 7: Comparison of our attackers. Randomness means it will produce different results in different trials. Trained means whether the parameters are tuned by a training set.

| Attack Method | Randomness | Trained | Time Complexity |
|---|---|---|---|
| ROCO-RA | ✓ | | $O(NK)$ |
| ROCO-OG | ✓ | | $O(BMK)$ |
| ROCO-SA | ✓ | ✓ | $O(NMK)$ |
| ROCO-RL | | ✓ | $O(B^2 K)$ |

Table 8: The ratio of budget K to average number of edges in different task. Each task has three datasets of different sizes, which is denoted by small, medium and large in the table.

| Tasks | Average Number of edges $|E|$ in three sizes | | | Budget $K$ | Ratio $K/|E|$ |
|---|---|---|---|---|---|
| | Small Size | Medium Size | Large Size | | |
| DAG | 468 | 930 | 1386 | 20 | $1.44\% \sim 4.27\%$ |
| ATSP | 400 | 2500 | 10000 | 10 | $0.10\% \sim 2.50\%$ |
| MC | 1260 | 1873 | 2497 | 10 | $0.40\% \sim 0.79\%$ |
| MCSCC | 4680 | 9567 | 16171 | 10 | $0.06\% \sim 0.21\%$ |

- **MCSCC**: ROCO-RA $N = 250$; ROCO-OG $B = 6$, $M = 36$; ROCO-SA $N = 25$, $M = 10$; ROCO-RL $B = 6$.

## J ANALYSIS OF ATTACK BUDGET

**Unnoticeable Property.** The perturbation to the problem instance is conducted by modifications to the graph structure. To get non-trivial hard instances around clean examples, the attack is limited within a given attack budget $K$, which means at most $K$ edges can be modified in a graph. Tbl. 1 compares our framework to classical works on adversarial attacks for images and graphs. Similar to the $\epsilon$-Perturb budget (Goodfellow et al., 2015) in image classification task and the number of edges modification (Dai et al., 2018) in graph classification task, we use the number of edges modification budget to satisfy the unnoticeable property. Tbl. 8 shows the ratio of budget $K$ to the average number of edges in these tasks. The low ratio shows that our modification to the original graph is minor. Apart from this, we also show the relation between the attack effect and the attack budget in the next paragraph.

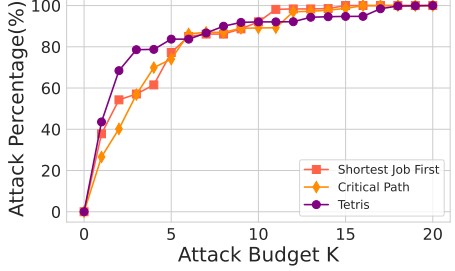

Figure 8: Relation between attack percentage to attack budget $K$ in the DAG task with problem size 50.

Figure 9: Relation between attack percentage to attack budget $K$ in the ATSP task with problem size 20.

**Sensitivity to the Budget $K$.** To evaluate sensitivity of the proposed method to the budget K across different tasks, we define attack percentage as the ratio of the performance loss in current edge budget to the the performance loss in the budget upper bound. For example, in ATSP task, the budget upper bound is 10 and we can will get $0\%$ attack percentage using 0 edge modification and $100\%$ using

Table 9: Comparison of different number of swapped node pairs $k$ on the "step by step improvement" modification for Nearest Neighbour algorithm on the dataset ATSP-20. Lower clean tour length represents better original performance, while lower attacked tour length and robustness ratio denote better robustness.

| Solver | Clean Tour Length ↓ | Attacked Tour Length ↓ | Robustness Ratio ↓ |
|---|---|---|---|
| Nearest Neighbour(Original) | 1.9354 | 2.0867 | 42.97% |
| Nearest Neighbour($k = 3$) | 1.7519 | 1.8838 | 29.07% |
| Nearest Neighbour($k = 5$) | 1.7258 | 1.8504 | 26.78% |
| Nearest Neighbour($k = 8$) | 1.7253 | 1.8462 | 26.50% |

Table 10: Comparison of different initial city number $s$ on the "different start state" modification for Furthest Insertion algorithm on the dataset ATSP-20.

| Solver | Clean Tour Length ↓ | Attacked Tour Length ↓ | Robustness Ratio ↓ |
|---|---|---|---|
| Furthest Insertion(Original) | 1.6092 | 1.6959 | 16.20% |
| Furthest Insertion($s = 3$) | 1.5602 | 1.6246 | 11.31% |
| Nearest Neighbour($s = 10$) | 1.5207 | 1.5774 | 8.07% |
| Nearest Neighbour($s = 20$) | 1.5005 | 1.5504 | 6.23% |

Table 11: Adversarial training(AT) for MatNet on dataset ATSP-20. MatNet-$X$ means we use $X$ fixed ATSP instances for training while MatNet-$X$-AT means we use $X$ hard cases generated from the clean instances by our framework.

| Data Type | MatNet-1000 | MatNet-1000-AT | MatNet-10000 | MatNet-10000-AT | MatNet |
|---|---|---|---|---|---|
| Clean instances | 1.4687 | 1.4691 | 1.4623 | 1.4626 | 1.4616 |
| Hard instances | 1.4761 | 1.4742 | 1.4737 | 1.4703 | 1.4708 |

10 edge modification. Fig. 8-11 show the relation between attack percentage to attack budget in different tasks. In Fig. 8, the three solvers all reach the high attack percentage using few budget. It may be more appropriate to reduce the budget because of marginal utility. Corresponding to this, it is possible for LKH3 to improve the attack performance as the budget rises shown in Fig. 9. Therefore, the sensitivity to budget $K$ can help set the appropriate attack budget in practice.

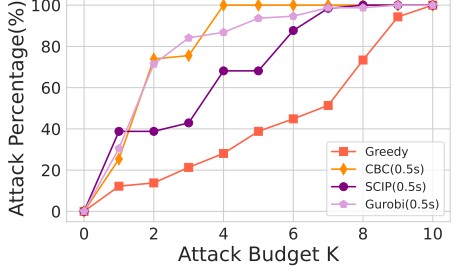

Figure 10: Relation between attack percentage to attack budget $K$ in the MC task with problem size 100-200.

Figure 11: Relation between attack percentage to attack budget $K$ in the MCSCC task with problem size 30-150-3k.

## K  ROBUSTNESS IMPROVEMENT

Apart from evaluating the robustness of the solvers, here we provide some preliminary yet positive results on improving solvers' robustness, while a full solution is indeed beyond the scope of this paper as we focus more on practical robustness evaluation.

Table 12: Hyperparameter tuning for Gurobi on MC(100-200). We focus on tuning two important hyperparameters: MIPFocus and Heuristics through Gurobi's API, which are described more detailedly in text.

| Hyperparameter | Hard Example Weight↑ | Clean Weight | ROCO-RL | Robustness Ratio↓ |
|---|---|---|---|---|
| MIPFocus=0, Heurstics=0.00 | 0.7414 | 0.7480 | 0.7418 | 0.83% |
| MIPFocus=0, Heurstics=0.05 | 0.7409 | 0.7491 | 0.7409 | 1.09% |
| MIPFocus=0, Heurstics=0.10 | 0.7386 | 0.7483 | 0.7383 | 1.34% |
| MIPFocus=1, Heurstics=0.00 | **0.7488** | 0.7478 | 0.7430 | **0.64%** |
| MIPFocus=1, Heurstics=0.05 | 0.7472 | 0.7490 | 0.7427 | 0.84% |
| MIPFocus=1, Heurstics=0.10 | 0.7455 | 0.7480 | 0.7410 | 0.94% |
| MIPFocus=2, Heurstics=0.00 | 0.7466 | 0.7460 | 0.7395 | 0.87% |
| MIPFocus=2, Heurstics=0.05 | 0.7448 | 0.7475 | 0.7398 | 1.03% |
| MIPFocus=2, Heurstics=0.10 | 0.7433 | 0.7466 | 0.7383 | 1.11% |

**Adversarial Training.** We perform adversarial training (AT) for MatNet (fixed training data) on dataset ATSP-20. Tbl. 11 presents our main results on AT. Specifically, MatNet-1000 denotes we use 1000 fixed ATSP instances to train the MatNet, while MatNet-1000-AT represents that we use 1000 hard instances generated by the ROCO-RL attacker to adversarially train the MatNet. The table shows that after AT, MatNet achieves obviously better results on the hard instances, at the cost of a bit performance degradation on the clean instances. However, AT cannot compensate for the insufficiency of training data, as we can see that the performance of MatNet-1000-AT is still worse than MatNet-10000 on the hard examples. The reason why we do not conduct AT on the original MatNet is that it trains for 2000 epochs, each with 10000 randomly generated ATSP instances, which is too large a number for us to generate adversarial samples. Notably, the results in the table are relatively close to each other which may introduce bias in the experiments. This implies that we may further explore other CO problems to test the effectiveness of AT, which we leave for the future work.

**LKH3-guided robustness improvement.** As we find from Tbl. 3 that LKH3 is the most robust solver for the ATSP problem, we try to adapt its randomization and optimization strategies to the Nearest Neighbour and the Furthest Insertion algorithm to promote their robustness.

**First, we develop Nearest Neighbour (Step by step improvement).** Instead of the well-designed optimization method k-opt (Lin, 1965) of LKH3, here we just greedily swap a pair of nodes on the solution obained by Nearest Neighbour for $k$ times. For each time we will enumerate all the node pairs and calculate the corresponding reduced tour length, then find the best pair to swap. We do experiments for different number of swaps $k$ on the dataset ATSP20. The robustness ratios (the lower the better) in Tbl. 9 are calculated w.r.t. the incumbent optimal solution. The incumbent optimal tour length is 1.4595 obtained by LKH3. We can find the clean tour length decreases, and the robustness increases with a larger value of $k$. Another interesting phenomenon is that the solver with $k = 5$ and $k = 8$ have similar clean tour lengths, while the latter is obviously more robust than the former. These results demonstrate that the optimization process does contribute greatly to the robustness of the solver.

**Second, we develop Furthest Insertion (Different start state).** The Furthest Insertion algorithm will be influenced by the initial city and generate different solutions accordingly. Therefore, we can set $s$ different cities as the initial city and choose the best solution as our result. We do experiments for different number of start states $s$ on the dataset ATSP20. From Tbl. 10, we can find that the clean tour length decreases, and the robustness increases with a larger value of $s$, which proves that randomization does contribute to the robustness of the solver.

**Hyperparameter tuning.** We use hard examples to tune the hyperparameters of Gurobi on the MC problem (problem size 100-200). We refer to Gurobi's Documentation in Gurobi Optimization (2020) and choose two important hyperparameters: MIPFocus and Heuristics, which are easily set by the python interface of Gurobi. Specifically, MIPFocus (integer, default 0, set between 0 and 3) allows us to modify our high-level solution strategy. By default, the Gurobi MIP solver strikes a balance between finding new feasible solutions and proving that the current solution is optimal. A higher MIPFocus will focus on proving optimality more. Heuristics (double, default 0.05, set between 0 and 1) determines the amount of time spent in MIP heuristics. Larger values produce more and better

feasible solutions, at a cost of slower progress in the best bound. We generate 100 hard examples from the testing dataset and tune the two hyperparameters using grid search for MIPFocus $\in [0, 1, 2]$ and Heuristics $\in [0, 0.05, 0.10]$.

As we can see from Tbl. 12, the robustness ratio (Clean Weight - ROCO-RL) / Clean Weight declines almost as the hard example weight (averaged over the cost value on 100 hard examples) increases. This shows that tuning the Gurobi hyperparameters is indeed useful to promote its robustness. Moreover, we can see that the hard example weight is the largest when MIPFocus=1 and Heuristics=0.0, which shows that Gurobi needs to pay more attention to narrowing the optimal bound to better deal with the hard examples.

## L  GENERALIZABILITY

One potential drawback of ROCO is its scalability. Since the training of the proposed RL strategy requires calling of the solver at each timestep, the training process indeed takes a relative long time (taking at most one day in our largest dataset setting, acceptable compared to common RL pipelines). However, the inference process is still efficient since we only need to take one forward pass with beam search. To resolve the concern that ROCO cannot scale to graphs in real-world scenarios, we provide additional results to show ROCO-RL's generalizability from small to large graphs.

Table 13: Evaluation results of ROCO-RL's generalizability on DAG. Each row denotes results on a testing size using models trained with different problem sizes, each column denotes testing a model trained on a certain problem size with different sizes of testing data. The robustness ratios are calculated w.r.t. the incumbent optimal solution. Lower robustness ratios denote better robustness.

| Solver | Shortest Job First | | | Critical Path | | | Tetris | | |
|---|---|---|---|---|---|---|---|---|---|
| Size: #job | 50 | 100 | 150 | 50 | 100 | 150 | 50 | 100 | 150 |
| 50 | 28.95% | 28.59% | 28.55% | 15.34% | 15.22% | 14.86% | 14.22% | 13.67% | 13.02% |
| 100 | 26.79% | 26.85% | 26.99% | 14.94% | 15.22% | 14.90% | 11.57% | 15.41% | 14.29% |
| 150 | 28.73% | 28.63% | 28.91% | 16.02% | 14.95% | 15.58% | 12.87% | 15.53% | 12.73% |
| 300 | 28.10% | 27.67% | 27.26% | 15.91% | 16.16% | 16.23% | 13.09% | 14.83% | 12.38% |

As we can see from Tbl. 13, the RL agents trained on small graphs can be used to modify problems with larger graph sizes, achieving great performance degradation for different solvers. These results strongly support that **ROCO can be trained on small graphs and then be used to measure robustness on large graphs.**

## M  FURTHER DISCUSSIONS

**Limitations.**  First of all, as our framework is designed for combinatorial optimization problems on graphs, it needs non-trivial reformulation to fit into CO problems cannot encoded by graphs. Secondly, the principle of designing our attack action is based on the graph modification to achieve no-worse optimum, which may not be applicable to all CO problems.

**Potential Negative Impacts.**  As we propose the first general framework to evaluate the robustness of combinatorial solvers on graphs, this may be used by unscrupulous people to attack different solvers, increasing the burden on engineers to cope with the malicious attacks.

**Outlook.**  Our paradigm opens up new space for further research, including the aspects: 1) Utilizing the hard cases generated by our framework to design defense models, such as adversarial training and hyperparameter tuning. 2) Developing a more general attack framework for CO problems that cannot be represented by graphs. 3) Besides our studied tasks, the attacking idea of loosening constraints and lowering costs is also generalizable on a wide range of CO problems such as Maximum Cut and Minimum Vertex Cover, which need further evaluation on their robustness.

