# OpenReview forum: "ROCO: A General Framework for Evaluating Robustness of Combinatorial Optimization Solvers on Graphs"
_ICLR.cc/2023/Conference — ICLR 2023 poster_

### Official Review · Reviewer_baVX · 2022-10-24

**Confidence:** 4
**Correctness:** 3
**Technical Novelty And Significance:** 2
**Empirical Novelty And Significance:** 3
**Recommendation:** 6

**Clarity, Quality, Novelty And Reproducibility:**

This paper presents an evaluation framework to measure the robustness of CO solvers. The problem setting is novel, while the technical novelty is a bit limited. The evaluation is conducted on various solvers and tasks with good quality.

**Strength And Weaknesses:**

Strength:

1. The problem, i.e., the robustness of CO solvers, is novel, interesting, and important.
2. The proposed framework evaluates different solvers and tasks extensively.

Weaknesses:

1. As an *evaluation* paper, the definition of robustness should be made clearer. I understand that the optimal solution is usually infeasible for NP-hard problems; however, a quantitative measure of robustness is still needed. For example, is the solver more robust with a lower (perturbed_solution - original_solution) absolute value or lower relative ratio (perturbed_solution - original_solution)/original_solution? And how to compare the robustness of two solvers if they get largely different original solutions?
2. Can the evaluation results lead to some insights about the robustness of different solvers, i.e., greedy solvers, heuristic solvers, neural network solvers, etc.?
3. Though the motivation to study the robustness of CO solvers is clear, more concrete examples/applications are needed to elaborate on why the attack model makes sense.
4. The novelty mainly comes from the problem definition, while the technical contribution is a little bit limited.

Minor: Page 5, the sentence above 3.2, edgesh -> edges

**Summary Of The Paper:**

This paper studies to measure the robustness of combinatorial solvers. The solver is defined as non-robust if the CO problem has relaxed constraints while the solver obtains a worse solution. This paper proposes to modify the corresponding graph structure for the CO problem to relax its constraints, and measure the performance of the solver on the modified graph compared to that on the original graph.

**Summary Of The Review:**

This paper studies a novel problem, i.e., the robustness of CO solvers, and provides an interesting framework that evaluates the robustness of various solvers on different tasks. More analysis of the evaluation results, elaboration of the applications of the attack model design, and more profound technical contributions are expected.

---

> ### Author Response · Authors · 2022-11-09
> **Response to Reviewer baVX(3/3)**
>
> >***Q3: More concrete examples/applications are needed to elaborate on why the attack model makes sense.***
>
> **R3:** Thank you for pointing out this insightful problem. We believe the following two examples can help you better understand how ROCO takes effect in real-world scenarios:
> - First, for the ATSP (Asymmetric TSP) problem, our attack model in the paper is to **choose certain edges and half their values**, which guarantees a no-worse theoretical optimum since the length of any path will maintain the same or even decrease. This kind of attack action widely exists in the real world since the paths between cities can change **from crowded to dredged** from time to time due to weather condition, traffic condition, etc. (e.g. a shorter distance, consistent with our halfing value action). In this scenario, the "attacker" is not a specific person but the ever-changing physical world. As proved by our experiments, this widely existing "attack" can degenerate the performance of ATSP solvers greatly.
>
> - Second, for the MCSCC problem (which is presented in Appendix B as a more complicated version of the MC problem), our attack model in the paper is to **add non-existing black edges that connect sets to black elements,** which leads to a no-worse optimum since we can possibly cover more black elements while not exceeding the white element threshold $k$. In fact, this problem is abstracted from a real-world problem called "Fraud Coverage": the black and white elements are both transactions (with weights standing for the amounts of transactions), with black $\mathcal B$ denoting *fraudulent* events, white $\mathcal W$ denoting *normal* transactions. To block fraud events, the bank system designs a series of rules $\mathcal S$ associated with certain set of transactions (i.e. elements), and the transactions (i.e. elements) covered by the selected rules $\mathcal S'\subseteq \mathcal S$ will be abandoned. In this sense, the objective for MCSCC can be interpreted as **selecting a subset of rules to maximize the coverage of fraudulent monetary values while not affecting no more than $k$ normal transactions**. And the "attack" action is to associate the rules with previously uncovered fraudulent transactions (i.e. black elements). This kind of attack is easy to implement by real-world attackers without much knowledge of the bank system. They can obtain information about transactions and rules from the bank's publicity content, then **report some fraudulent transactions** (i.e. black elements) to make them covered by additional rules. As proved by our experiments, this kind of "attack" can degenerate the performance of a wide range of CO solvers including Gurobi.
>
> Besides, we have shown some successful attacked examples in Fig.4-6 in the original paper.
>
> >***Q4: The novelty mainly comes from the problem definition, while the technical contribution is a little bit limited.***
>
> **R4:** While it is true that the paper's focus is on defining the first general framework to evaluate the robustness of CO solvers on graphs, we believe that **our technical contribution is more than a simple combination of GNNs, RL, and heuristic search.**
> - First, the insight of "no worse cost attack" is an important technical contribution as it paves the way for follow-up works to measure the robustness of CO solvers without requiring the optimal value.
> - Second, we provide inspiring examples on how we can successfully "attack" the problem instances w.r.t. specific CO problems such as DAG, TSP, and Maximum Coverage. We also propose different and effective pruning techniques to narrow the action space for each problem. In a word, we develop effective attack methods **with clear physical meaning (see R3)** for different CO problems by well combining exsting techniques as well as proposing problem-specific modifications.
> - Third, we investigate the factors associated with the robustness of solvers and pioneer to utilize the hard samples discovered by our attackers to improve solvers' robustness. The initial results are presented in Appendix K, which we believe can encourage future works to dig deeper in this area.
>
> Also thank you for pointing out the typos, which have been fixed in the updated version.
>
> ### References
> [1] On Evaluating Adversarial Robustness, arXiv 2019.
>
> [2] On Collective Robustness of Bagging against Data Poisoning, ICML 2022.
>
> [3] MIPLIB, http://softlib.rice.edu/miplib.html.
>
> [4] Matrix Encoding Networks for Neural Combinatorial Optimization, NeurIPS 2021.
>
> [5] Computer solutions of the traveling salesman problem, Bell Labs Technical Journal 1965.
>
> [6] It’s Not What Machines Can Learn, It’s What We Cannot Teach, ICML 2020.

---

> ### Author Response · Authors · 2022-11-09
> **Response to Reviewer baVX(2/3)**
>
> >***Q2: Can the evaluation results lead to more insights about the robustness of different solvers?***
>
> **R2:** Thank you for pointing out the oversight in our original paper. Here we provide more in-depth analysis of solvers' robustness.
> - **Neural Solvers.** From Tbl. 3, we can see that "MatNet(fixed)" which is trained with 10,000 fixed ATSP instances is always less robust than "MatNet" that is trained with randomly generated ATSP instances during each epoch. It proves that huge amount of i.i.d. training instances promotes both performance and the robustness of the model. As [6] points out, learning-based solvers for CO cannot be trained sufficiently due to the theoretical limit of data generalization. Thus, the training paradigm with large unlabeled data [4] can be a good choice for learning-based CO solvers. Besides, feeding hard samples discovered by ROCO to "MatNet(fixed)" to conduct adversarial training may also promote its robustness.
> - **Heuristic Solvers.** In the ATSP task, we find the LKH3 is the most robust method among heuristic solvers. We attribute the phenomenon to two aspcet of LKH3: 1. Different initial states. LKH3 will take 4 runs with diffenent initial states in our setting. 2. Step by step improvement. Specifically, LKH3 uses k-opt [5] to iteratively reorder and optimize the result tour sequence. The introduction of randomization and optimization should contribute to the strong robustness of LKH3. Inspired by LKH3, we can use these two strategies to improve the robustness of suboptimal solvers such as Nearest Neighbor and Furtherest Insertion.
> - **MILP Solvers.** In the Maximum Coverage problem, our attack model to MC problem is equivalent to modifying the objective coefficients of the original ILP model of MC problem, which might incur numerical issues for ILP solver such as unstable matrix factorization. Since we can access the source code of CBC solver and SCIP solver respectively, we find that the SCIP solver **is equipped with more modules improving numerical stability of solving process, such as scaling, presolving and primal heuristics, etc.** SCIP solver empirically behaves more robustness over many benchmarks (e.g. MIPLIB [3]) than CBC. Besides, SCIP also benefits from implementing more classes of **branching policy and cutting planes** that aim to improve the dual bound, than CBC. The experimental results in Tbl. 4 are in line with our expectation for the tested ILP solvers, where CBC solver performs worse than SCIP solver in terms of solution quality and robustness. Compared to the open-sourced solvers, the commercial solver Gurobi performs undoubtedly better in the perspective of both solution quality and robustness. We can also tune its hyperparameters through the python API to achieve better robustness.
>
> We have added these analysis in the newly submitted version. Besides, we have also conducted more experiments to verify the hypothesis proposed above (improve solvers' robustness), you can refer to the general response and Appendix K for more details.

---

> ### Author Response · Authors · 2022-11-09
> **Response to Reviewer baVX(1/3)**
>
> Thank you for your time, comments, and acknowledgements for the empirical novelty and significance of our work. Here we provide additional experimental results and clarifications to address your concerns:
> >***Q1: The definition of robustness should be made clearer, i.e. a quantitative measure of robustness is needed.***
>
> **R1:** Thanks for pointing out our oversight in the text. Instead of using the performance degradation gap after attacked or the ratio normalized w.r.t. the original cost value, we use the raw value to stand for robustness in the paper. Typically, as different solvers can produce largely different original solution for a same problem, the metric of gap would favor the bad solvers as they could just perform equally bad on the original and modified instance. On the contrary, solvers with good performance may degenerate greatly, giving a large performance gap and therefore a high metric value. So this metric is unfair for robustness comparison between different solvers. Next, we explain why we use the raw value in the tables.
>
> A consensus in the field of robustness is that **the robustness definition should be against the attacker's objective [1][2].** For example, in [1] (which studies adversarial robustness) the attacker attempts to achieve the maximum classification loss on the test set, and the robustness is defined to be the worst case loss for a given perturbation budget. In [2] (which studies certified robustness) the attacker tries to maximize the number of misclassified samples on the test set, and the robustness is denoted by the certified accuracy after attacked (rather than the accuracy gap). In our paper, the attacker's target is given by Eq.3 in the original paper. Similar to [1][2], we **therefore define a solver's robustness as the worst case cost value after attacked on the test set (equivalent to the average raw value):**
> $$
> \text{robustness}=-\mathbb E_{Q\sim\mathcal{D}}[\max_{\tilde{Q}}c(f_\theta(\tilde{Q}),\tilde{Q})]\\
> s.t. \tilde{Q}=g(f_\theta,Q), c(\tilde{\mathbf{x}}^*,\tilde{Q})\leq c(\tilde{\mathbf{x}}^*,Q)
> $$
> where $\mathcal{D}$ denotes the test set. We add the negative value sign so that a higher value denotes better robustness. The logic of our robustness measure has been updated in the newly submitted version.
>
> Finally, **we also propose a more intuitive robustness metric in terms of ratios.** To avoid the problem in the first paragraph and associate the ratio with the attacker's objective, we use the incumbent optimal solution (i.e. the best solution we can find among all solvers) cost to define the robustness:
> $$
> \text{robustness} = -\frac{\text{attacked cost}-\text{incumbent optimal cost}}{\text{incumbent optimal cost}}
> $$
> where we omit the expectation and constraints for simplicity. As the metric is just the attacked cost value normalized w.r.t. the shared incumbent optimal cost, it is still against the attacker's objective. We have added this metric in the newly submitted version.

---

> ### Comment · Reviewer_baVX · 2022-12-08
> **post-rebuttal**
>
> Thanks for the authors' detailed response. The definition of robustness is made clearer and more formal now. I raise the score from 5 to 6.

---

### Official Review · Reviewer_oJSV · 2022-10-25

**Confidence:** 4
**Correctness:** 3
**Technical Novelty And Significance:** 3
**Empirical Novelty And Significance:** 2
**Recommendation:** 5

**Clarity, Quality, Novelty And Reproducibility:**

The originality and quality of work are okay but not good enough for the standards of ICLR. There are many experiments in this paper as it evaluates 3 solvers on 4 problems using 5 variants of the proposed metric. Yet, I would expect to see more in-depth analyses of these results and some follow-up experiments investigating the robustness of why and when a model/solver tends to be more robust, which is also important for telling the usefulness of a proposed metric and why the proposed metric is evaluating the real robustness of the model. Also, the proposed evaluation metric needs problem-specific designs of how to modify the graph structure and may blindly favor a solver that performs averagely on different problem instances.
The presentation of the work could be better by providing a). rigorous definitions of the terminology used and b) better visualization of the robustness comparison.

**Strength And Weaknesses:**

Strength:
-

[+] This work tackles an important domain: how to evaluate the robustness of solvers for Combinatorial Optimization problems.

[+] The proposed evaluation metric does not need the optimal value of the underlying CO problem instance and does not require the solver to be differentiable.

---
Weakness:
-

[-] I do not see much insight into why and when a solver would be more robust on a problem than other solvers or other problems from the current experimental results. This part is important in order to tell the usefulness of ROCO and why it is evaluating the real robustness of the model.

[-] The proposed evaluation metric needs problem-specific designs of how to modify the graph structure (e.g., add or remove edges, change edge length, etc)in the attack model.

[-] Sometimes it may be easier to degenerate the performance of good solvers than the bad ones especially when the good one is close to the optimal but the bad one is close to some baseline value, in this case, I think the proposed metric would favor the bad one as it could just perform equally bad on the original and modified instance.

---

Minor comments:
- It would be better to compare the robustness using ratios rather than raw numbers for all tasks in the main paper.
- The presentation of the work could be better if some terminology could be defined (earlier) such as "attacker" and "no worse optimal". costs"

**Summary Of The Paper:**

This work propose a new evaluation metric for evaluating the robustness of solvers on combinatorial optimization (CO) problems. The proposed evaluation metric is agnostic to optimal values, which may be hard to obtain in real world situations. As the proposed metric requires another problem instance with better optimal objective value but "harder" to solve, this work also proposed a RL-based approach, to modify the structure of the input graph which represents the underlying CO problem. It then evaluate the robustness of three combinatorial solvers (traditional solvers, learning based solver and specific MILP solver like Gurobi) using the proposed metric with the proposed approach on four combinatorial tasks: Directed Acyclic Graph Scheduling, Asymmetric Traveling Salesman Problem, Maximum Coverage, and Maximum Coverage with Separate Coverage Constraint.

Overall, the proposed evaluation metric has some novelty over other existing metrics, but it needs problem-specific designs of how to modify the graph structure (e.g., add or remove edges, change edge length, etc). Also, the training of the proposed RL strategy requires call of the solver at each step, which may not be practical for real-world scenarios where the graphs may have thousands (or even more) of nodes. More important, I think this work would have more important contributions if it could utilize ROCO to provide more insights on what make one solver more robust than other solvers or what makes one solver more robust on a specific problem than other problems. Such follow-up experiments and analysis are important to tell the usefulness of ROCO and why it is evaluating the real robustness of the model.





**Summary Of The Review:**

This paper studied the robustness of a combinatorial solver as a black box regardless of its differentiability. It developed a robustness metric that does not need the optimal solution by using another instance with no-worse optimal cost guarantee. It uses an RL-based approach to find such instances given some hand-designed relaxation rules given a specific problem. In the experimental section, this work conducted extensive experiments on 14 different combinations of solvers and variants of robust metrics. Yet, it lacks in-depth analyses of why and when a solver is more robust under this metric as well as how the proposed metric is evaluating the real robustness of the model.

---

> ### Author Response · Authors · 2022-11-09
> **Response to Reviewer oJSV(4/4)**
>
> >***Q4: The proposed metric seems to favor the bad solvers that give some baseline value.***
>
> **R4:** The problem indeed exists if we use the performance degradation gap to represent a solver's robustness. Therefore, we use the raw cost value after attacked rather than the gap or the gap normalized w.r.t. the original cost value to denote solvers' robustness in the original paper. Here we explain the reason of our choice.
>
> A consensus in the field of robustness is that, **the robustness definition should be against the attacker's objective [6][7].** For example, in [6] (which studies adversarial robustness) the attacker attempts to achieve the maximum classification loss on the test set, and the robustness is defined to be the worst case loss for a given perturbation budget. In [7] (which studies certified robustness) the attacker tries to maximize the number of misclassified samples on the test set, and the robustness is denoted by the certified accuracy after attacked (rather than the accuracy gap). In our paper, the attacker's target is given by Eq.3 in the original paper. Similar to [6][7], we **therefore define a solver's robustness as the worst case cost value after attacked on the test set (equivalent to the average raw value):**
> $$
> \text{robustness}=-\mathbb E_{Q\sim\mathcal{D}}[\max_{\tilde{Q}}c(f_\theta(\tilde{Q}),\tilde{Q})]\\
> s.t. \tilde{Q}=g(f_\theta,Q), c(\tilde{\mathbf{x}}^*,\tilde{Q})\leq c(\tilde{\mathbf{x}}^*,Q)
> $$
> where $\mathcal{D}$ denotes the test set. We add the negative value sign so that a higher value denotes better robustness. The logic of our robustness measure has been updated in the newly submitted version.
>
> With this in mind, we can also find some interesting results from the experimental results. For the Critical Path and Tetris solvers on the DAG problem with size 100, and the CBC and SCIP solvers on the MC problem with size 200-400, we have:
> | DAG Solvers              | Critical Path | Tetris | MC Solvers | CBC (1.5s) | SCIP (1.5s) |
> | ------------------- | ------------- | ------ | ------------------- | ------------- | ------ |
> | Original Problem    | 1.6018        | 1.5186 |Original Problem    | 1.4248        | 1.3994 |
> | RL Attacked Problem | 1.7498        | 1.7526 |RL Attacked Problem | 1.0418        | 1.3536 |
>
> The table shows that the Tetris solver after attacked is less robust than Critical Path, while it performs significantly better than Critical Path on the original test set. Similarly, CBC is less robust than SCIP though performs better on the original problems.
>
> >***Q5: It would be better to compare the robustness using ratios rather than raw numbers.***
>
> **R5:** Thank you for pointing out the problem. Typically, the robustness is not necessarily a ratio (e.g. [6] adopts worst case loss). However, here **we also propose a more intuitive robustness metric in terms of ratios.** To avoid the problem in Q4 and associate the ratio with the attacker's objective, we use the incumbent optimal solution (i.e. the best solution we can find among all solvers) cost to define the robustness:
> $$
> \text{robustness} = -\frac{\text{attacked cost}-\text{incumbent optimal cost}}{\text{incumbent optimal cost}}
> $$
> where we omit the expectation and constraints for simplicity. As the metric is just the attacked cost value normalized w.r.t. the shared incumbent optimal cost, it is still against the attacker's objective. We have added this metric in the newly submitted version.
>
> >***Q6: Better to provide rigorous definitions of the terminology used (earlier) such as "attacker" and "no worse optimal costs".***
>
> **R6:** Thank you for pointing out the problems of our presentation. We have modify the corresponding parts in the introduction and methodology section (highlighted in blue in the updated version) to make the paper more easy to read. Specifically, "attackers" are perturbation models that modify the CO problems to get hard problem instances (where the solvers perform poorly) with no worse optimal costs. Successively, "no worse optimal costs" means that the optimal solution of the modified problem instance gives better objective value than the original problem.
>
> ### References
> [1] Generalization of Neural Combinatorial Solvers Through the Lens of Adversarial Robustness, ICLR 2022.
>
> [2] Matrix Encoding Networks for Neural Combinatorial Optimization, NeurIPS 2021.
>
> [3] Learning Scheduling Algorithms for Data Processing Clusters, SIGCOMM 2019.
>
> [4] Coloring Big Graphs with AlphaGoZero, arXiv 2019.
>
> [5] NP-completeness, https://en.wikipedia.org/wiki/NP-completeness.
>
> [6] On Evaluating Adversarial Robustness, arXiv 2019.
>
> [7] On Collective Robustness of Bagging against Data Poisoning, ICML 2022.
>
> [8] MIPLIB, http://softlib.rice.edu/miplib.html.
>
> [9] Computer solutions of the traveling salesman problem, Bell Labs Technical Journal 1965.
>
> [10] It’s Not What Machines Can Learn, It’s What We Cannot Teach, ICML 2020.

---

> ### Author Response · Authors · 2022-11-09
> **Response to Reviewer oJSV(3/4)**
>
> We tune the hyperparameters of Gurobi based on the hard samples discovered by ROCO on the MC problem (problem size 100-200), whose results are shown in Tbl.12 in Appendix K. We refer to Gurobi's Documentation and choose two important hyperparameters: MIPFocus and Heuristics, which are easily set by the python interface of Gurobi. Specifically, MIPFocus (integer, default 0, set between 0 and 3) allows us to modify our high-level solution strategy. By default, the Gurobi MIP solver strikes a balance between finding new feasible solutions and proving that the current solution is optimal. A higher MIPFocus will focus on proving optimality more. Heuristics (double, default 0.05, set between 0 and 1) determines the amount of time spent in MIP heuristics. Larger values produce more and better feasible solutions, at a cost of slower progress in the best bound. We generate 100 hard examples from the testing dataset and tune the two hyperparameters using grid search for MIPFocus $\in$ [0, 1, 2] and Heuristics $\in$ [0, 0.05, 0.10]. As we can see from the table, the robustness ratio $\frac{\text{Clean Weight} - \text{ROCO-RL}}{\text{Clean Weight}}$ declines almost as the hard example weight (averaged over the cost value on 100 hard examples) increases. This shows that tuning the Gurobi hyperparameters is indeed useful to promote its robustness. Moreover, we can see that the hard example weight is the largest when MIPFocus=1 and Heuristics=0.0, which shows that Gurobi needs to pay more attention to narrowing the optimal bound to better deal with the hard examples.
>
> These results have all been added in the newly submitted version.
>
>
> >***Q3: The proposed evaluation metric needs problem-specific designs of how to modify the graph structure.***
>
> **R3:** Admittedly, our ROCO needs problem-specific designs of how to modify the graph structure. However, we do not regard it as a critical problem for the following reasons:
> - As CO problems greatly differ from each other in terms of objectives and constraints, existing works often tend to deal with them individually. For example, [2] studies about TSP problems, [3] focues on scheduling problems, while [4] is proposed to deal with graph coloring. Besides, a similar work [1] to our paper which needs optimal value also proposes problem-specific design for measuring robustness of neural solvers on SAT and TSP.
> - Besides, we claim that our graph modification method is more general and easier to come up with than that in [1]. It can be generally divided into two types: loosening the constraints and lowering the cost, which we have discussed detailedly in Appendix F. Following this general principle, we can easily find more examples of attacks on different CO problems:
>     - Minimum Vertex Cover: The attack model can add edges between unconnected nodes. The optimal vertex set in the clean instance can still cover all the vertices in the attacked instance. A better solution may exist because some vertices can cover more, replacing the role of other nodes.  It is a format of the strategy loosening the constraints in our paper.
>     - Maximum Cut: The attack model can add the value of the edges. Then the original optimal cut will stay the same or increase, making "no-worse optimum". It is a format of the strategy lowering the costs in our paper.
>   - Maximum Independent Set: The attack model can remove edges between connected nodes. Therefore, the original optimal set will still be available while a better solution may exist because some of the edges are removed and some nodes will not be influenced by the selecting nodes. It is a format of the strategy loosening the constraints in our paper.
> - Last but not least, as a well known theory that NP-hard problems can be reduced to each other in polynomial time [5], here we also provide a **potentially generic** version of ROCO. That is, we can first transform the input CO problem into an equivalent TSP instance, for example. Then we can attack the TSP problem instead (which has been studied in the paper), and transform it back to the original problem type. Finally, we re-run the solver on the attacked problem and calculate rewards for the RL agent.

---

> ### Author Response · Authors · 2022-11-09
> **Response to Reviewer oJSV(2/4)**
>
> - **Heuristic Solvers.** In the ATSP task, we find the LKH3 is the most robust method among heuristic solvers. We attribute the phenomenon to two aspcet of LKH3: 1. Different initial states. LKH3 will take 4 runs with diffenent initial states in our setting. 2. Step by step improvement. Specifically, LKH3 uses k-opt [9] to iteratively reorder and optimize the result tour sequence. The introduction of randomization and optimization should contribute to the strong robustness of LKH3. Inspired by LKH3, we can use these two strategies to improve the robustness of suboptimal solvers such as Nearest Neighbor and Furtherest Insertion.
>     - **First, we develop Nearest Neighbour (Step by step improvement).** Instead of the well-designed optimization method k-opt [9] of LKH3, here we just greedily swap a pair of nodes on the solution obained by Nearest Neighbour for $k$ times. For each time we will enumerate all the node pairs and calculate the corresponding reduced tour length, then find the best pair to swap. We do experiments for different number of swaps $k$ on the dataset ATSP20. The robustness ratios (the lower the better) in the following table are calculated by the metric we proposed in R5. The incumbent optimal tour length is 1.4595 obtained by LKH3. We can find the clean tour length decreases, and the robustness increases with a larger value of $k$, which proves the correctness of our hypothesis. Another interesting phenomenon is that the solver with $k=5$ and $k=8$ have similar clean tour lengths, while the latter is obviously more robust than the former. This supports that the optimization process does contribute greatly to the robustness of the solver.
>       | Solver     | Clean Tour Length $\downarrow$ | Attacked Tour Length $\downarrow$ | Robustness Ratio $\downarrow$ |
>       | ---------- | ---------- | ------------- | ---------------- |
>       | Nearest Neighbour(Original) | 1.9354     | 2.0867        | 42.97%           |
>       | Nearest Neighbour($k = 3$)  | 1.7519     | 1.8838        | 29.07%            |
>       | Nearest Neighbour($k = 5$)  | 1.7258     | 1.8504        | 26.78%            |
>       | Nearest Neighbour($k = 8$)  | 1.7253     | 1.8462        | 26.50%            |
>
>   - **Second, we develop Furthest Insertion (Different start state).** The Furthest Insertion algorithm will be influenced by the initial city and generate different solutions accordingly. Therefore, we can set $s$ different cities as the initial city and choose the best solution as our result. We do experiments for different number of start states $s$ on the dataset ATSP20. From the following table, we can find that the clean tour length decreases, and the robustness increases with a larger value of $s$, which proves that randomization does contributes to the robustness of the solver.
>
>     | Solver    | Clean Tour Length $\downarrow$ | Attacked Tour Length $\downarrow$ | Robustness Ratio $\downarrow$ |
>     | ---------- | ---------- | ------------- | ---------------- |
>     | Furthest Insertion(Original) | 1.6092     | 1.6959        | 16.20%            |
>     | Furthest Insertion($s = 3$)  | 1.5602     | 1.6246        | 11.31%            |
>     | Furthest Insertion($s = 10$) | 1.5207     | 1.5774        | 8.07%            |
>     | Furthest Insertion($s = 20$) | 1.5005     | 1.5504        | 6.23%            |
>
> - **MILP Solvers.** In the Maximum Coverage (MC) problem, our attack model to MC problem is equivalent to modifying the objective coefficients of the original ILP model of MC problem, which might incur numerical issues for ILP solver such as unstable matrix factorization. Since we can access the source code of CBC solver and SCIP solver respectively, we find that the SCIP solver **is equipped with more modules improving numerical stability of solving process, such as scaling, presolving and primal heuristics, etc.** SCIP solver empirically behaves more robustness over many benchmarks (e.g. MIPLIB [8]) than CBC. Besides, SCIP also benefits from implementing more classes of **branching policy and cutting planes** that aim to improve the dual bound, than CBC. The experimental results in Tbl. 4 are in line with our expectation for the tested ILP solvers, where CBC solver performs worse than SCIP solver in terms of solution quality and robustness. Compared to the open-sourced solvers, the commercial solver Gurobi performs undoubtedly better in the perspective of both solution quality and robustness. We can also tune its hyperparameters through the python API to achieve better robustness.

---

> ### Author Response · Authors · 2022-11-09
> **Response to Reviewer oJSV(1/4)**
>
> Thanks for your time and constructive feedback. We are pleased that you acknowledged the technical novelty and significance of our work. Here we provide more experimental results and insights to address your concerns:
> >***Q1: The training of the proposed RL strategy requires call of the solver at each step, which may not scale to real-world scenarios where graphs have thousands of nodes.***
>
> **R1:** It is a fact that the training process of our RL agent is relatively slow (taking at most one day in our largest dataset setting, acceptable compared to common RL pipelines), since we need to re-run the solver at each timestep. However, the inference process is still efficient since we only need to take one forward pass with beam search.
> To resolve your concern that ROCO cannot scale to graphs in real-world scenarios, we provide additional results to show ROCO-RL's generalizability from small to large graphs as such the training scalability issue can be mitigated.
>
> | Solver      |     | SJF |     |     | CP |     |     | Tetris |     |
> | ------------ | --- | ------------------ | --- | --- | ------------- | --- | --- | ------ | --- |
> | Problem Size | 50  | 100                | 150 | 50  | 100           | 150 | 50  | 100    | 150 |
> | 50           | 28.95%  | 28.59%     | 28.55%  | 15.34% | 15.22% | 14.86% | 14.22% | 13.67% | 13.02%    |
> | 100          | 26.79%    |  26.85%      | 26.99% | 14.94% | 15.22% | 14.90%    | 11.57% | 15.41%       |  14.29%   |
> | 150          | 28.73%    |  28.63%      | 28.91% | 16.02% | 14.95% | 15.58%    | 12.87% | 15.53%       |  12.73%   |
> | 300      | 28.10% | 27.67%    | 27.26% | 15.91%  | 16.16%               |  16.23%   | 13.09%    |   14.83%   | 12.38%    |
>
> where each row denotes results on a testing size using models trained with different problem sizes, and each column denotes testing a model trained on a certain problem size with different sizes of testing data. The ratios are calculated w.r.t. the metric we proposed in R5. As we can see, the RL agents trained on small graphs can be used to modify problems with larger graph sizes, achieving great performance degradation for different solvers. These results strongly support that **ROCO can be trained on small graphs and then be used to measure robustness on large graphs.** We have added these results to the appendix.
>
> >***Q2: More insight into why and when a solver would be more robust on a problem than other solvers or other problems.***
>
> **R2:** Thank you for pointing out our oversight in the original paper. Here we provide more in-depth study of solvers' robustness.
> - **Neural Solvers.** From Tbl. 3, we can see that "MatNet(fixed)" which is trained with 10,000 fixed ATSP instances is always less robust than "MatNet" that is trained with randomly generated ATSP instances during each epoch. It proves that huge amount of i.i.d. training instances promotes both performance and the robustness of the model. As [10] points out, learning-based solvers for CO cannot be trained sufficiently due to the theoretical limit of data generalization. Thus, the training paradigm with large unlabeled data [2] can be a good choice for learning-based CO solvers. Besides, feeding hard samples discovered by ROCO to "MatNet(fixed)" to conduct adversarial training (AT) may also promote its robustness.
> Typically, we perform adversarial training for MatNet(fixed) on dataset ATSP-20. The results are shown in Tbl. 11 in the Appendix K. The table shows that after AT, MatNet achieves obviously better results on the hard instances, at the cost of a bit performance degradation on the clean instances. However, AT cannot compensate for the insufficiency of training data, as we can see that the performance of MatNet-1000-AT is still worse than MatNet-10000 on the hard examples.

---

> ### Author Response · Authors · 2022-12-08
> **To reviewer oJSV: Looking forward to your reply**
>
> Dear reviewer oJSV,
>
> Thank you for your valuable guidance and give us the opportunity to improve our paper based your valuable comments (please see the updated pdf and our response with more dicussion and experimental results).
>
> We are pleased that reviewer bavx raised his/her socre who once had the similar concerns and questions like you, on the robustness metric and insights of the solvers. So we also humbly ask for your feedback  at your convenience such that we may still have the chance to discuss with you and chance to improve our final version before the close of rebuttal window.

---

### Official Review · Reviewer_2fgV · 2022-10-25

**Confidence:** 3
**Correctness:** 3
**Technical Novelty And Significance:** 3
**Empirical Novelty And Significance:** 3
**Recommendation:** 6

**Clarity, Quality, Novelty And Reproducibility:**

The writeup quality is decent. There are some typos.
Presentation could be improved if the goal for robustness were mentioned upfront.

**Strength And Weaknesses:**

S1: The new metric does not need to know the optimum.
S2: It treats the solver as black-box, so it also works with non-differentiable combinatorial optimization problems.
S3: The RL-based approach works well in practice.

W1: Why is this a good notion of robustness? It's not clear what robustness is supposed to achieve.
W2: A robustness metric should perhaps produce a number between 0 and 1. While one can simply normalize with respect to the original solution, there is no discussion of the advantages or disadvantages.
W3: There could be more discussion about relative robustness of the different solvers. For example, CBC performs poorly in terms of robustness on max cover instances. It also performs poorly in terms of quality compared to the other solvers. Is there more to read from this?

**Summary Of The Paper:**

This manuscript proposes a new method (called ROCO) to study robustness of algorithms/solvers for combinatorial optimization problems. It does not need the optimal solution, nor does it require a differnetiable solver.

To avoid the need for knowing the optimal solution, they modify the input instance in such a way that the optimal cannot become worse. For example, you can reduce edge weights or relax certain constraints. Then they re-run the solver on the new instance and expect that the solution will not produce worse objective values. How much worse it gets determines the level of non-robustness.

Then they get to the problem of finding these "hard" instances for the solver. Their reinforcement learning-based method outperforms a few baselines (random edge selection, "optimum guided" via beam search, and simulated annealing). In their empirical study, they look at a few combinatorial optimization problems like task scheduling and max coverage.

**Summary Of The Review:**

Decent paper with interesting practical results, though the empirical section and the initial motivation sections could be expanded with more insight.

---

> ### Author Response · Authors · 2022-11-09
> **Response to Reviewer 2fgV(2/2)**
>
> >***Q3: More discussion about relative robustness of the different solvers.***
>
> **R3:** Thank you for pointing out the oversight in our original paper. Here we provide more in-depth analysis of solvers' robustness.
> - **Neural Solvers.** From Tbl. 3, we can see that "MatNet(fixed)" which is trained with 10,000 fixed ATSP instances is always less robust than "MatNet" that is trained with randomly generated ATSP instances during each epoch. It proves that huge amount of i.i.d. training instances promotes both performance and the robustness of the model. As [3] points out, learning-based solvers for CO cannot be trained sufficiently due to the theoretical limit of data generalization. Thus, the training paradigm with large unlabeled data [6] can be a good choice for learning-based CO solvers. Besides, feeding hard samples discovered by ROCO to "MatNet(fixed)" to conduct adversarial training may also promote its robustness.
> - **Heuristic Solvers.** In the ATSP task, we find the LKH3 is the most robust method among heuristic solvers. We attribute the phenomenon to two aspcet of LKH3: 1. Different initial states. LKH3 will take 4 runs with diffenent initial states in our setting. 2. Step by step improvement. Specifically, LKH3 uses k-opt [5] to iteratively reorder and optimize the result tour sequence. The introduction of randomization and optimization should contribute to the strong robustness of LKH3. Inspired by LKH3, we can use these two strategies to improve the robustness of suboptimal solvers such as Nearest Neighbor and Furtherest Insertion.
> - **MILP Solvers.** In the Maximum Coverage problem, our attack model to MC problem is equivalent to modifying the objective coefficients of the original ILP model of MC problem, which might incur numerical issues for ILP solver such as unstable matrix factorization. Since we can access the source code of CBC solver and SCIP solver respectively, we find that the SCIP solver **is equipped with more modules improving numerical stability of solving process, such as scaling, presolving and primal heuristics, etc.** SCIP solver empirically behaves more robustness over many benchmarks (e.g. MIPLIB [4]) than CBC. Besides, SCIP also benefits from implementing more classes of **branching policy and cutting planes** that aim to improve the dual bound, than CBC. The experimental results in Tbl. 4 are in line with the expectation for the tested ILP solvers, where CBC solver performs worse than SCIP solver in terms of solution quality and robustness. Compared to the open-sourced solvers, the commercial solver Gurobi performs undoubtedly better in the perspective of both solution quality and robustness. We can also tune its hyperparameters through the python API to achieve better robustness.
>
> We have added these analysis in the newly submitted version. Besides, we have also conducted more experiments to verify the hypothesis proposed above (improve solvers' robustness), you can refer to the general response and Appendix K for more details.
>
> ### References
> [1] On Evaluating Adversarial Robustness, arXiv 2019.
>
> [2] On Collective Robustness of Bagging against Data Poisoning, ICML 2022.
>
> [3] It's Not What Machines Can Learn, It's What We Cannot Teach, ICML 2020.
>
> [4] MIPLIB, http://softlib.rice.edu/miplib.html.
>
> [5] Computer solutions of the traveling salesman problem, Bell Labs Technical Journal 1965.
>
> [6] Matrix Encoding Networks for Neural Combinatorial Optimization, NeurIPS 2021.

---

> ### Author Response · Authors · 2022-11-09
> **Response to Reviewer 2fgV(1/2)**
>
> Thank you for your time and valuable comments. We are glad that you like the paper's novelty, significance, and presentation. And for your concerns, our responses are as follows:
> >***Q1: It's not clear what robustness is supposed to achieve. Presentation could be improved if the goal for robustness were mentioned upfront.***
>
> **R1:** Thank you for pointing out this to improve our presentation. In the filed of computer vision, a series of works [1][2] have validated that simple perturbation on pixels can greatly harm the classifiers' performance. In the context of CO, similarly, [3] proposes the "no free lunch" theorem that we cannot teach a perfect solver to predict satisfying results for all input problems, i.e. there must exist some CO problems (i.e. the hard instances) around the fitting data distribution where the trained solver performs poorly. This has also been validated by our experiments that solvers' performance may degenerate drastically under a limited number of modifications to the original CO problem. So in general, **a solver's robustness is its capability to perform stably w.r.t. the perturbation on CO problems drawn from certain distribution (i.e. the hard instances).** We have added this interpretation in the introduction in the updated version.
>
> Notably, these kinds of perturbation or "attack" are widely existing in the real world. For example, in the real world the paths between cities can change from crowded to dredged very frequently, corresponding to our attack to the ATSP problem. We also present another example in R3 to reviewer baVX (omitted here because it is not the focus). These examples further illustrate the significance of our robustness. We have also proposed some initial results for improving solvers' robustness in Appendix K. Furthermore, you can refer to our general response of "robustness metric" if you're confused about our robustness measurement.
>
> >***Q2: Discussion of the pros and cons of normalizing w.r.t. the original solution to obtain a robustness value between 0 and 1.***
>
> **R2:** Thank you for pointing out this valuable problem.
> - First, we explain why we do not use the relative ratio $\frac{\text{attacked cost}-\text{original cost}}{\text{original cost}}$ to denote robustness (the lower the better) in the original paper, which seems to be a reasonable alternative since we cannot get the optimal solution. However, as different solvers can produce largely different original solution for a same problem, this metric would favor the bad solvers as they could just perform equally bad on the original and modified instance. On the contrary, solvers with good performance may degenerate greatly, giving a large performance gap and therefore a high metric value. So this metric is unfair for robustness comparison between different solvers.
> - Next, we explain why we use the raw value in the table. Naturally and widely recognized, **the robustness definition should be against the attacker's objective [1][2].** For example, in [1] (which studies adversarial robustness) the attacker attempts to achieve the maximum classification loss on the test set, and the robustness is defined to be the worst case loss for a given perturbation budget. In [2] (which studies certified robustness) the attacker tries to maximize the number of misclassified samples on the test set, and the robustness is denoted by the certified accuracy after attacked (rather than the accuracy gap). In our paper, the attacker's target is given by Eq.3 in the original paper. Similar to [1][2], we **therefore define a solver's robustness as the worst case cost value after attacked on the test set (equivalent to the average raw value):**
> $$
> \text{robustness}=-\mathbb E_{Q\sim\mathcal{D}}[\max_{\tilde{Q}}c(f_\theta(\tilde{Q}),\tilde{Q})]\\
> s.t. \tilde{Q}=g(f_\theta,Q), c(\tilde{\mathbf{x}}^*,\tilde{Q})\leq c(\tilde{\mathbf{x}}^*,Q)
> $$
> where $\mathcal{D}$ denotes the test set. We add the negative value sign so that a higher value denotes better robustness. The logic of our robustness measure has been updated in the newly submitted version.
> - It is important to note that robustness is not necessarily a ratio (e.g. [1] adopts worst case loss). However, here **we also propose a more intuitive robustness metric in terms of ratios.** To avoid the problem in the first point and associate the ratio with the attacker's objective, we use the incumbent optimal solution (i.e. the best solution we can find among all solvers) cost to define the robustness:
> $$
> \text{robustness} = -\frac{\text{attacked cost}-\text{incumbent optimal cost}}{\text{incumbent optimal cost}}
> $$
> where we omit the expectation and constraints for simplicity. As the metric is just the attacked cost value normalized w.r.t. the shared incumbent optimal cost, it is still against the attacker's objective. We have added this metric in the newly submitted version.

---

### Official Review · Reviewer_Agfm · 2022-10-27

**Confidence:** 3
**Correctness:** 3
**Technical Novelty And Significance:** 3
**Empirical Novelty And Significance:** 3
**Recommendation:** 8

**Clarity, Quality, Novelty And Reproducibility:**

The paper is clearly written in general. The work seems new and I think the originality is non-trivial. One of the technical contributions is to define a new success criterion of an attack. The problem is well motivated and relevant.

**Strength And Weaknesses:**

Strength:
- A reasonable definition of the success of an attack for combinatorial optimization solvers
- Several implementations of attacking are proposed and evaluated in experiments


Weakness:
- The problems are restricted to graph problems so far

The key idea of the paper is to propose a reasonable criterion of the success of an attack. The previous work needs an optimal solution, which is intractable in general. The current work does not have such a disadvantage and becomes more practical.



Minor technical issue:
- In Definition 1, there is no definition of \tilde{x^*}.

**Summary Of The Paper:**

The paper proposes a framework and algorithms for verifying the robustness of combinatorial optimization solvers for graph problems. A key idea is to define a new criterion of successful attacks to the solver, defined without using the optimal solutions, whereas the previous work requires them. The paper then proposes some methods to implement the attacks. Among them, one attack is implemented based on reinforcement learning. Experimental results show the advantages of the RL-based attack against other proposed methods and the random naive method.

**Summary Of The Review:**

The paper proposes a reasonable definition of the success of an attack for combinatorial optimization solvers over graphs.

---

> ### Author Response · Authors · 2022-11-09
> **Response to Reviewer Agfm**
>
> Thanks for your positive comments and favorable assessment. Our responses are as follows:
> >***Q1: The studied CO problems are restricted to graph problems so far.***
>
> **R1:** The subject of this paper is delineated in the title: we study the robustness of CO solvers on graphs. That is, **we are upfront about acknowledging this as our limitation, but also believe that it is not a critical problem.** Here we state our points of view:
> - First, as a well known theorem that NP-hard problems can be reduced to each other in polynomial time [3], we claim that graph is a general representation for CO problems. For example, to measure a solver's robustness on a CO problem without explicit graph structure, we can transform it to an equivalent TSP problem (which can be represented as a graph) and attack it instead. The TSP problem after attacked can then be transformed to the original problem class, on which we re-run the solver to calculate the rewards.
> - Second, as the theorem above supports our framework design from a high-level, we do not need to resort to it frequently since a large number of CO problems can be represented as graphs inherently, e.g. Travelling Salesman Problem(TSP), Vehicular Routing Problem(VRP), Maximum Cut, Minimum Vertex Cover(MVC), Maximum independent set(MIS), Maximum Coverage(MC), Graph Coloring(GC)...... Besides, the constraints of "no worse optimum" are also generalizable on other CO problems, for example:
>     - Minimum Vertex Cover: The attack model can add edges between unconnected nodes. The optimal vertex set in the clean instance can still cover all the vertices in the attacked instance. A better solution may exist because some vertices can cover more, replacing the role of other nodes.  It is a format of the strategy loosening the constraints in our paper.
>     - Maximum Cut: The attack model can add the value of the edges. Then the original optimal cut will stay the same or increase, making "no-worse optimum". It is a format of the strategy lowering the costs in our paper.
>   - Maximum Independent Set: The attack model can remove edges between connected nodes. Therefore, the original optimal set will still be available while a better solution may exist because some of the edges are removed and some nodes will not be influenced by the selecting nodes. It is a format of the strategy loosening the constraints in our paper.
> - Last but not least, ROCO is compatible with other forms of problem encoders. For example, MatNet [1] encodes TSP problems using multi-head attention, while Ptr-Net [2] resorts to RNNs instead. Replacing GNNs in ROCO with these kinds of encoders can help it encode other input formats such as matrices [1]. A step further, the attacker is also not limited to do modification on graphs. For example, we can half the value in the distance matrix to attack a TSP problem while compromising to the "no worse optimum" constraint.
>
> >***Q2: In Definition 1, there is no definition of $\tilde{\mathbf{x}}^\*$.***
>
> **R2:** Thank you for pointing out the problem. Typically, $\mathbf{\tilde{x}}^*$ denotes the optimal solution of the attacked problem $\tilde{Q}$. We have fixed this issue in the updated version.
>
> ### References
> [1] Matrix Encoding Networks for Neural Combinatorial Optimization, NeurIPS 2021.
>
> [2] Pointer Networks, NeurIPS 2015.
>
> [3] NP-completeness, https://en.wikipedia.org/wiki/NP-completeness.

---

### Author Response · Authors · 2022-11-09
**General Response(3/3)**

The robustness ratio (the lower the better) is calculated w.r.t. the incumbent optimal solution. The incumbent optimal tour length is 1.4595 obtained by LKH3. We can find the clean tour length decreases, and the robustness increases with a larger value of $k$, which proves the correctness of our hypothesis. Another interesting phenomenon is that the solver with $k=5$ and $k=8$ have similar clean tour lengths, while the latter is obviously more robust than the former. This supports that the optimization process does contribute greatly to the robustness of the solver.

--**Furthest Insertion (Different start state).** The Furthest Insertion algorithm will be influenced by the initial city and generate different solutions accordingly. Therefore, we can set $s$ different cities as the initial city and choose the best solution as our result. We do experiments for different number of start states $s$ on the dataset ATSP20.

  | Solver    | Clean Tour Length $\downarrow$ | Attacked Tour Length $\downarrow$ | Robustness Ratio $\downarrow$ |
  | ---------- | ---------- | ------------- | ---------------- |
  | Furthest Insertion(Original) | 1.6092     | 1.6959        | 16.20%            |
  | Furthest Insertion($s = 3$)  | 1.5602     | 1.6246        | 11.31%            |
  | Furthest Insertion($s = 10$) | 1.5207     | 1.5774        | 8.07%            |
  | Furthest Insertion($s = 20$) | 1.5005     | 1.5504        | 6.23%            |

  The incumbent optimal tour length is 1.4595 obtained by LKH3. We can find the clean tour length decreases, and the robustness increases with a larger value of $s$, which proves that randomization does contribute to the robustness of the solver.

--**Hyperparameter tuning.** We tune the hyperparameters of Gurobi based on the hard samples discovered by ROCO on the MC problem (problem size 100-200), whose results are shown in Tbl.12 in Appendix K. We refer to Gurobi's Documentation in [7] and choose two important hyperparameters: MIPFocus and Heuristics, which are easily set by the python interface of Gurobi. Specifically, MIPFocus (integer, default 0, set between 0 and 3) allows us to modify our high-level solution strategy. By default, the Gurobi MIP solver strikes a balance between finding new feasible solutions and proving that the current solution is optimal. A higher MIPFocus will focus on proving optimality more. Heuristics (double, default 0.05, set between 0 and 1) determines the amount of time spent in MIP heuristics. Larger values produce more and better feasible solutions, at a cost of slower progress in the best bound. We generate 100 hard examples from the testing dataset and tune the two hyperparameters using grid search for MIPFocus $\in$ [0, 1, 2] and Heuristics $\in$ [0, 0.05, 0.10]. As we can see from the table, the robustness ratio $\frac{\text{Clean Weight} - \text{ROCO-RL}}{\text{Clean Weight}}$ declines almost as the hard example weight (averaged over the cost value on 100 hard examples) increases. This shows that tuning the Gurobi hyperparameters is indeed useful to promote its robustness. Moreover, we can see that the hard example weight is the largest when MIPFocus=1 and Heuristics=0.0, which shows that Gurobi needs to pay more attention to narrowing the optimal bound to better deal with the hard examples.

These results have also been added in the newly submitted version.


We provide supplementary experimental results and detailed answers to all the questions raised by the reviewers in the following individual responses. Besides, we have also revised the paper w.r.t. the suggestions of the reviewers, which are highlighted in blue in the newly submitted version.

### References
[1] On Evaluating Adversarial Robustness, arXiv 2019.

[2] On Collective Robustness of Bagging against Data Poisoning, ICML 2022.

[3] Computer solutions of the traveling salesman problem, Bell Labs Technical Journal 1965.

[4] It's Not What Machines Can Learn, It's What We Cannot Teach, ICML 2020.

[5] Matrix Encoding Networks for Neural Combinatorial Optimization, NeurIPS 2021.

[6] MIPLIB, http://softlib.rice.edu/miplib.html.

[7] Gurobi, http://www.gurobi.com.

---

### Author Response · Authors · 2022-11-09
**General Response(2/3)**

**2. More insights on the robustness of solvers.** As reviewers encourage us to give more insights on the robustness of CO solvers, we provide additional experimental results and in-depth analysis here.
- **Our added interpretation of robustness behavior of different solvers.**

    --**Neural Solvers.** From Tbl. 3, we can see that "MatNet(fixed)" which is trained with 10,000 fixed ATSP instances is always less robust than "MatNet" that is trained with randomly generated ATSP instances during each epoch. It proves that huge amount of i.i.d. training instances promotes both performance and the robustness of the model. As [4] points out, learning-based solvers for CO cannot be trained sufficiently due to the theoretical limit of data generalization. Thus, the training paradigm with large unlabeled data [5] can be a good choice for learning-based CO solvers. Besides, feeding hard samples discovered by ROCO to "MatNet(fixed)" to conduct adversarial training may also promote its robustness.

    --**Heuristic Solvers.** In ATSP task, we find the LKH3 is the most robust method among heuristic solvers. We attribute the phenomenon to two aspcets of LKH3: 1. Different initial states. LKH3 will take 4 runs with diffenent initial states in our setting. 2. Step by step improvement. Specifically, LKH3 uses k-opt [3] to iteratively reorder and optimize the result tour sequence. The introduction of randomization and optimization should contribute to the strong robustness of LKH3. Inspired by LKH3, we can use these two strategies to improve the robustness of suboptimal solvers such as Nearest Neighbor and Furtherest Insertion.

    --**MILP Solvers.** In the Maximum Coverage problem, our attack model to MC problem is equivalent to modifying the objective coefficients of the original ILP model of MC problem, which might incur numerical issues for ILP solver such as unstable matrix factorization. Since we can access the source code of CBC solver and SCIP solver respectively, we find that the SCIP solver **is equipped with more modules improving numerical stability of solving process, such as scaling, presolving and primal heuristics, etc.** SCIP solver empirically behaves more robustness over many benchmarks (e.g. MIPLIB [6]) than CBC. Besides, SCIP also benefits from implementing more classes of **branching policy and cutting planes** that aim to improve the dual bound, than CBC. The experimental results in Tbl.4 are in line with the expectation for the tested ILP solvers, where CBC solver performs worse than SCIP solver in terms of both solution quality and robustness. Compared to the open-sourced solvers, the commercial solver Gurobi performs undoubtedly better in the perspective of both solution quality and robustness. We can also tune its hyperparameters through the python API to achieve better robustness.
    We have added these analysis in the newly submitted version.

- **Our added new results for improving robustness based on our insights.**

     --**Adversarial Training.** We perform adversarial training (AT) for MatNet (fixed training data) on dataset ATSP-20. The results are shown in Tbl. 11 in the Appendix K. Specifically, MatNet-1000 denotes we use 1000 fixed ATSP instances to train the MatNet, while MatNet-1000-AT represents that we use 1000 hard instances generated by the ROCO-RL attacker to adversarially train the MatNet. The table shows that after AT, MatNet achieves obviously better results on the hard instances, at the cost of a bit performance degradation on the clean instances. However, AT cannot compensate for the insufficiency of training data, as we can see that the performance of MatNet-1000-AT is still worse than MatNet-10000 on the hard examples. The reason why we do not conduct AT on the original MatNet is that it trains for 2000 epochs, each with 10000 randomly generated ATSP instances, which is too large a number for us to generate adversarial samples.

    --**Nearest Neighbour (Step by step improvement).** Instead of the well-designed optimization method k-opt [3] of LKH3, here we just greedily swap a pair of nodes on the solution obained by Nearest Neighbour for $k$ times. For each time we will enumerate all the node pairs and calculate the corresponding reduced tour length, then find the best pair to swap. We do experiments for different number of swaps $k$ on the dataset ATSP20.

  | Solver     | Clean Tour Length $\downarrow$ | Attacked Tour Length $\downarrow$ | Robustness Ratio $\downarrow$ |
  | ---------- | ---------- | ------------- | ---------------- |
  | Nearest Neighbour(Original) | 1.9354     | 2.0867        | 42.97%           |
  | Nearest Neighbour($k = 3$)  | 1.7519     | 1.8838        | 29.07%            |
  | Nearest Neighbour($k = 5$)  | 1.7258     | 1.8504        | 26.78%            |
  | Nearest Neighbour($k = 8$)  | 1.7253     | 1.8462        | 26.50%            |

---

### Author Response · Authors · 2022-11-09
**General Response(1/3)**

Dear Area Chair and Reviewers,
We would like to express our sincere thanks to the reviewers' for their time and valuable suggestions. Overall, we are encouraged that most reviewers acknowledged the paper's significance (Agfm, 2fgV, oJSV, baVX), techical novelty (Agfm, 2fgV, oJSV), and positive experimental results (Agfm, 2fgV, baVX).

Foremost, we try to address and clarify some common concerns:

**1. Robustness metric.** As required by the reviewers, here we provide a quantitative metric to measure the robustness of CO solvers as for their direct comparisons, and explain its rationale.
- First, we explain the defect of using the relative ratio $\frac{\text{attacked cost}-\text{original cost}}{\text{original cost}}$ as the  robustness metric (the lower the better). In fact, as it is often impossible to obtain the optimal value of CO problems, reviewer baVX proposes this simple normalization metric for robustness. However, as different solvers can produce distinctive solutions for the original problem instance, it will encounter the issue pointed out by reviewer oJSV: this metric would possibly favor the bad solvers as they could just perform equally bad on the original and modified instance. On the contrary, solvers with good performance may degenerate greatly, giving a large performance gap and therefore a high metric value. So this metric is unfair for robustness comparison between different solvers.
- Second, we give the reason why we choose **the raw cost value after attacked** for robustness in our original paper. Naturally and widely recognized, **the robustness definition should be against the attacker's objective [1][2].** For example, in [1] (which studies adversarial robustness) the attacker attempts to achieve the maximum classification loss on the test set, and the robustness is defined to be the worst case loss for a given perturbation budget. In [2] (which studies certified robustness) the attacker tries to maximize the number of misclassified samples on the test set, and the robustness is denoted by the certified accuracy after attacked (rather than the accuracy gap). So in our paper, the attacker's target is given by Eq. 3 in our original paper:
$$
\max_{\tilde{Q}} c(f_\theta(\tilde{Q}),\tilde{Q})-c(f_\theta(Q),Q) \\
s.t. \tilde{Q}=g(f_\theta,Q), c(\tilde{\mathbf{x}}^*,\tilde{Q})\leq c(\tilde{\mathbf{x}}^*,Q)
$$
where the attacker tries to maximize the cost after attacked under the "no worse optimal cost" constraint. Similar to [1][2], we therefore **define a solver's robustness as the worst case cost value after attacked on the test set (equivalent to the average raw value):**
$$
\text{robustness}=-\mathbb E_{Q\sim\mathcal{D}}[\max_{\tilde{Q}}c(f_\theta(\tilde{Q}),\tilde{Q})]\\
s.t. \tilde{Q}=g(f_\theta,Q), c(\tilde{\mathbf{x}}^*,\tilde{Q})\leq c(\tilde{\mathbf{x}}^*,Q)
$$
where $\mathcal{D}$ denotes the test set. We add the negative value sign so that a higher value denotes better robustness. The logic of our robustness measure has been updated in the newly submitted version.
- Finally, as required by reviewers 2fgV and oJSV, **we propose a more intuitive metric of robustness in terms of ratios.** However, it is important to note that robustness is not necessarily a ratio (e.g. [1] adopts worst case loss). To avoid the problem in the first point and associate the ratio with the attacker's objective, we use the incumbent optimal solution (i.e. the best solution we can find among all solvers) cost to define the robustness:
$$
\text{robustness} = -\frac{\text{attacked cost}-\text{incumbent optimal cost}}{\text{incumbent optimal cost}}
$$
where we omit the expectation and constraints for simplicity. As the metric is just the attacked cost value normalized w.r.t. the shared incumbent optimal cost, it is still against the attacker's objective. We have added this metric in the newly submitted pdf.

---

### Author Response · Authors · 2022-11-16
**Inquiry for post-rebuttal comments**

Dear Reviewers:

Thank you again for your wisdom and valuable comments. We have provided experimental or complete explanations for all the questions. Since the discussion period is approaching its end, we would be glad to hear from you whether our rebuttal has addressed your concerns. Feel free to comment on our rebuttal if you have further questions and considerations.

---

### Author Response · Authors · 2022-11-18
**To all**

Since the pdf update period is approaching its end, we would be glad to hear from you whether our rebuttal has addressed your concerns, and is there anything needed to be added. Feel free to comment on our rebuttal if you have further questions and considerations.

---

### Author Response · Authors · 2022-12-08
**Inquiry for post-rebuttal discussion**

Dear Area Chair,

Would you mind to start the discussion for our paper (it not done yet)?  It would be appreciated if we had the chance to response your further questions after your internal discussion.

---

### Decision · Program_Chairs · 2023-01-20

**Decision:**

Accept: poster

**Justification For Why Not Higher Score:**

During the discussions among the reviewers, it seemed that the reviewers are happy with the contributions, but were not willing to give a higher score that what they had already given.

**Justification For Why Not Lower Score:**

It's a clear accept.

**Metareview: Summary, Strengths And Weaknesses:**

The paper proposes a framework and algorithms for verifying the robustness of combinatorial optimization solvers for graph problems. All the reviewers are in favor of accepting the paper, and believe that the paper is an interesting contribution (this was also clear in the discussion among the reviewers). I recommend the authors to incorporate the comments from the reviewers in the revised version (please see the updated reviews). Also, please add the new experimental results to the paper.

**Note From Pc:**

if the above contains the word "oral" or "spotlight" please see: "oral" presentation means -> notable-top-5% and "spotlight" means -> notable-top-25%. As stated in our emails, we are disassociating presentation type from AC recommendations